# Concept, Design, Initial Tests and Prototype of Customized Upper Limb Prosthesis

**Corina Radu (Frenț)** [1], **Maria Magdalena Roșu** [2], **Lucian Matei** [3], **Liviu Marian Ungureanu** [2] **and Mihaiela Iliescu** [1,*]

1   Institute of Solid Mechanics, Romanian Academy, Constantin Mille 15, 010141 Bucharest, Romania; corinagabrielaradu@gmail.com
2   Faculty of Industrial Engineering and Robotics, University POLITEHNICA of Bucharest, Splaiul Independentei 313, 060042 Bucharest, Romania; magdalena.rosu@upb.ro (M.M.R.); liviu.ungureanu@upb.ro (L.M.U.)
3   Faculty of Mechanics, University of Craiova, Alexandru Ioan Cuza 13, 200585 Craiova, Romania; matei.lucian@ucv.ro
*   Correspondence: mihaiela.iliescu@imsar.ro

**Abstract:** This paper presents aspects of the concept and design of prostheses for the upper limb. The objective of this research is that of prototyping a customized prosthesis, with EMG signals that initiate the motion. The prosthesis' fingers' motions (as well as that of its hand and forearm parts) are driven by micro-motors, and assisted by the individualized command and control system. The software and hardware tandem concept of this mechatronic system enables complex motion (in the horizontal and vertical plane) with accurate trajectory and different set rules (gripping pressure, object temperature, acceleration towards the object). One important idea is regarding customization via reverse engineering techniques. Due to this, the dimensions and appearance (geometric characteristics) of the prosthesis would look like the human hand itself. The trajectories and motions of the fingers, thumbs, and joints have been studied by kinematic analysis with the matrix–vector method aided by Matlab. The concept and design of the mechanical parts allow for complex finger motion—rotational motion in two planes. The command and control system is embedded, and data received from the sensors are processed by a micro-controller for managing micro-motor control. Preliminary testing of the sensors and micro-motors on a small platform, Arduino, was performed. Prototyping of the mechanical components has been a challenge because of the high accuracy needed for the geometric precision of the parts. Several techniques of rapid prototyping were considered, but only DLP (digital light processing) proved to be the right one.

**Keywords:** prosthesis; reverse engineering; 3D prototyping; embedded command and control system; sensors

## 1. Introduction

The upper limb is an essential organ for any person due to its ability to complete a lot of actions with a higher level of dexterity. This is how, by using their upper limbs, people can achieve tasks such as grasping, holding, picking, and lifting more efficiently than other living creatures can. According to the World Health Organization report on disability [1], about "15% of the global population—over a billion people—lives with some form of disability, of whom 2–4% experience significant difficulties in functioning".

According to Kim et al. [2], cited by [3], "the ratio of upper-extremity to lower-extremity amputations is 1:2.2" and the "persons with upper-extremity amputations suffer from frustration and difficulty during the rehabilitation process. The main factors causing loss of the upper limbs are accidents followed by general diseases and injuries, and also in some cases for diseases and tumours, the amputation is a way of stopping the spread of the disease to the rest of the body [3]".



Many of these people "suffer from frustration and difficulty during the rehabilitation process and experience paresthesia due to the loss of delicate movement by the hands, complicated tactile senses of the upper extremities, and proprioceptive sensory functions [3]" and all of these people require assistive technologies such as prostheses, wheelchairs, etc.

Considering the crucial functions of hands, the life of people who are missing hands is complicated, and, for this reason, it is very difficult to adapt to situations in which something that used to be done naturally is now done within limits imposed by this disability. Upper limb prostheses (ULP) are generally classified into two categories based on their functionality: passive prostheses and active prostheses [4], cited by [5]. A good solution for artificial hands is myoelectric prostheses, which are currently being developed. With these types of prostheses, the myoelectric signals of muscle contractions are ultimately transformed into functions desired by the users through the use of surface electrodes [3]. It is a 'myoelectric control system' for controlling the myoelectric prosthesis [6], cited by [3]. However, the design, the control structure, and the manufacturing process are difficult, and moreover, the cost is a very important element to consider in all of these phases [3,5]. An adaptable hand with flexibility, dexterity, and load carrying capacity analogous to the human hand seems to be the ideal objective for prosthetic application [7].

Prosthetics research has contributed to the development of very complicated hand prostheses but their structures, with programmable control systems, are not always user-friendly. This makes them very difficult in operation, and the people at whom they are aimed do not use them in the usual way [7–10].

Thus, it is necessary to develop intelligent prostheses that can be easily adapted to the requirements of each individual they are intended for. For the development of such products, researchers must consider the following basic systems: mechanical, actuators and motors, sensors, and command and control [10].

According to [11], for designing a hand prosthesis, the project team needs to understand the mechanics of mechanisms such as gears, levers, and points of mechanical advantage, and electromechanical design such as switches, DC motors, and electronics. Mechatronics is the new word used to describe this "marriage" of mechanical and electronic engineering. However, for applying the concepts of mechatronics in bioengineering, it is necessary to take into account the principles of how the human body functions.

Electromyography (EMG) [12] is used to evaluate and record the electrical activity produced by the muscles of human body. Recently, it has been used in the rehabilitation of patients with amputations in the form of robotic prostheses, it is a tool used to evaluate and classify the different movements of body, so that the robotic mechanism can effectively imitate the motions of human limbs. In [13], mathematical algorithms tested in prototypes of intracortical neuromotor prostheses have been developed. These closed-loop algorithms [13] offer visual access (neurally derived cursor trajectories), and mechanical access (as in the stimulation of muscles via implanted electrodes) to any number of output devices.

The upper limb prostheses should be customized, complex, high-performance, and able to ensure a good life for the person wearing it. There are many hand prostheses available on this special "market", but efficient prostheses are expensive and are not affordable for all of the people who might need them.

This paper presents aspects of the concept and design of a customized prosthesis for the upper limb. The software and hardware tandem concept of this mechatronic system enables complex motion (in the horizontal and vertical planes) with accurate trajectory and different set rules (gripping pressure, object temperature, acceleration toward the object). The command and control system is embedded, and the data received from the sensors are processed by a micro-controller for managing the control of the micro-motors. The customized prosthesis is intended to be light and user-friendly, is estimated to have medium manufacturing costs (a few thousand Euros), and would be reliable, with little maintenance required.

## 2. Materials and Methods

One of the first steps in the conceptualization of the customized upper limb prosthesis is that of the kinematic analysis of its finger mechanism. In order to perform this, some basic aspects of the mechanism theory have been considered as follows [14–22]:

- Link—is a rigid body.
- Joint—is a permanent contact between two links.
- Planar motion—is the motion "in which all points belonging to a link move in a plane known as the plane of motion, while simultaneously the link is free to rotate about an axis perpendicular to the plane of motion".
- Degree of freedom of a rigid body—is the number of independent movements it has.

Kinematic constraints are "constraints between rigid bodies that result in the decrease of the degrees of freedom of the rigid body system".

An Assur group is a set of kinematic chains developed by Leonid Assur whose degree of freedom is zero. A complex structure can be constructed by extending the Assur kinematic chain. The complex linkage mechanism can be regarded as originating from some Assur kinematic chains

Kinematic analysis is the motion analysis aimed at determining the positions, speed, and acceleration distributions of each element, previously knowing their constructive characteristics and the relative movement of the active link elements.

A dyad is an Assur group made of two parts, three joints, and two outputs (Figure 1a,b).

A triad is an Assur group made of four parts, six joints, and three outputs (Figure 1c).

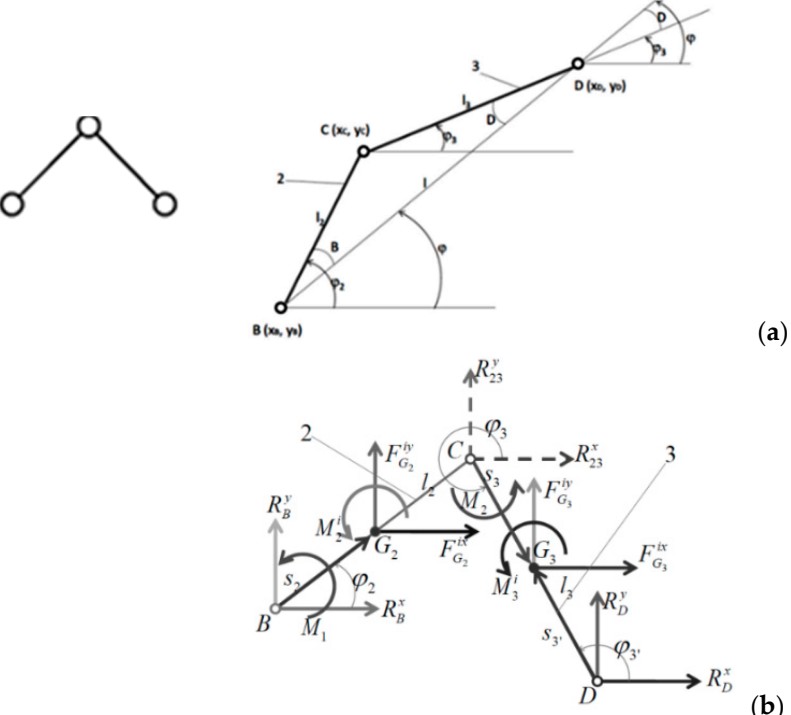

**(a)**

**(b)**

**Figure 1.** *Cont.*

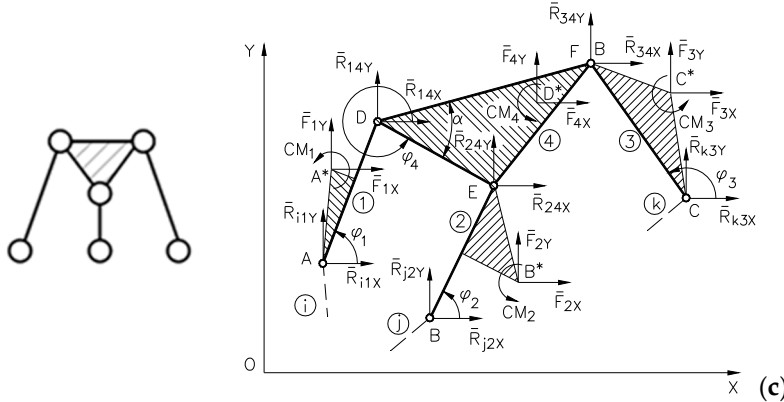

**Figure 1.** Assur group types—dyad and triad: (**a**) kinematic scheme for the dyad (RRR) [21], (**b**) kinetostatic scheme for the dyad (RRR) [21], and (**c**) kinematic and kinetostatic schemes for the triad (RR-RR-RR) [22] where: $R$ is rotation, $\overline{F}$ is the force vector (components along axes), and $\overline{R}$ is the joint reaction force vector (components along axes).

For the mechanism of the designed prosthesis, the kinematic analysis was done by the matrix–vector method [23,24], which is simple, intuitive, and fits for open spatial contours. For its index finger mechanism, the kinematic scheme is shown in Figure 2. The final Hartenberg–Denavit trihedral, $T_1$, $T_2$, ..., $T_{i+1}$, is associated with the fingertips. The trihedral versos are $\overline{n}_i$, $\overline{e}_{i+1} \times \overline{n}_i$.

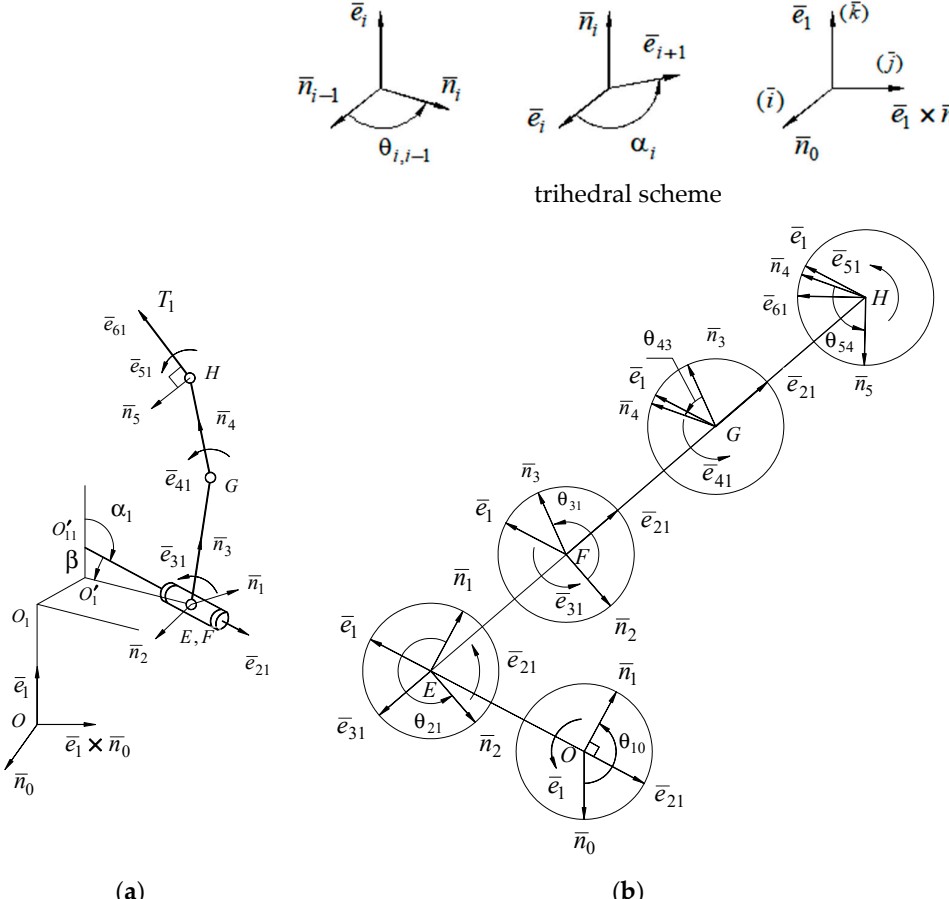

**Figure 2.** Schemes of the designed index finger mechanism: (**a**) kinematic scheme, (**b**) versorial scheme.

The main steps of the calculus required for the kinematic analysis of the index finger mechanism are presented next. The simplified structure of the mechanism is a serial open kinematic chain made of dyads. The base trihedral, $\overline{n}_0, \overline{e}_1 \times \overline{n}_0, \overline{e}_1$, overlaps the orthogonal reference trihedral Oxyz with the $\overline{i}, \overline{j}, \overline{k}$ versors. The position vector of the $T_1$ customized point, $\overline{r}_{T_1}$, is given by relation (1):

$$\overline{r}_{T_1} = s_1\overline{e_1} + a_1\overline{n_1} + s_2\overline{e_2} + a_2\overline{n_2} + s_3\overline{e_3} + a_3\overline{n_3} + s_4\overline{e_4} + a_4\overline{n_4} + s_5\overline{e_5} + a_5\overline{n_5} + s_6\overline{e_6} + a_6\overline{n_6} \quad (1)$$

where:

- $s_1$ represents the distance between normal versors, $\overline{n}_{i-1}$ and $\overline{n}_i$ (axial displacement), with a positive sign along the $\overline{e}_i$ axis;
- $a_i$ is the distance between the $\overline{e}_i$ and $\overline{e}_{i+1}$ axes, positive sign;
- $\theta_{i, i-1}$ is the angle between the $\overline{n}_{i-1}\overline{n}_{i-1}$ and $\overline{n}_i$ versors, positive sign along $\overline{e}_i$ axis;
- $\alpha_i$ is the angle between the $\overline{e}_i$ and $\overline{e}_{i+1}$ versors (crossing angle), positive sign along $\overline{n}_i$ axis.

In order to simplify the transformation matrix elements' notation, the symbols are: $\theta_1 = \theta_{10}, \theta_2 = \theta_{20}$, and $\theta_3 = \theta_{30}$.

Coordinates transformation of a point, from $T_i$ trihedral to $T_{i+1}$ trihedral, is:

$$T_{i+1} = A_i \cdot T_i \quad (2)$$

where:

$$A_i = \begin{bmatrix} \cos\theta_i & \sin\theta_i & 0 \\ -\sin\theta_i\cdot\cos\alpha_i & \cos\theta_i\cdot\cos\alpha_i & \sin\alpha_i \\ \sin\theta_i\cdot\sin\alpha_i & -\cos\alpha_i\cdot\sin\alpha_i & \cos\alpha_i \end{bmatrix} \quad (3)$$

is the transformation matrix.

Consequently, there are evidenced the relations that follows:

$$T_{i+1} = A_i\cdot T_i = A_i(A_{i-1}\cdot T_{i-1}) = A_iA_{i-1}(A_{i-2}\cdot T_{i-2}) = A_iA_{i-1}\ldots A_2A_1T_1 \quad (4)$$

and

$$\overline{n}_i = D_i\overline{n}_0 + E_i\overline{e}_1 \times \overline{n}_0 + F_i\overline{e}_1 \quad (5)$$

$$\overline{e}_{i+1} = A_{i+1}\overline{n}_0 + B_{i+1}\overline{e}_1 \times \overline{n}_0 + C_{i+1}\overline{e}_1 \quad (6)$$

where $A_{i+1}, B_{i+1}, C_{1+1}, D_i, E_i, F_i, K_i, L_i, M_i$ represent the component elements of the transformation matrix,

$$\overline{A}_i = A_iA_{i-1}\cdot\cdots\cdot A_2A_1 = \begin{bmatrix} D_i & E_i & F_i \\ K_i & L_i & M_i \\ A_{i+1} & B_{i+1} & C_{i+1} \end{bmatrix} \quad (7)$$

Based on all the above, the projections of the position vector, $\overline{r}_{T1}$, on the Oxyz trihedral are:

$$\begin{aligned} X_T &= a_1\cdot D_1 + a_2\cdot D_2 + \cdots + a_5\cdot D_5 + s_2\cdot A_2 + s_3\cdot A_3 + \cdots + s_6\cdot A_6 \\ Y_T &= a_1\cdot E_1 + a_2\cdot E_2 + \cdots + a_5\cdot E_5 + s_2\cdot B_2 + s_3\cdot B_3 + \cdots + s_6\cdot B_6 \\ Z_T &= a_2\cdot F_2 + a_3\cdot F_3 + \cdots + a_5\cdot F_5 + s_1 + s_2\cdot C_2 + s_3\cdot C_3 + \cdots + s_6\cdot C_6 \end{aligned} \quad (8)$$

or, written as a matrix, it turns into:

$$\begin{bmatrix} X_T & Y_T & Z_T \end{bmatrix}^T = P\cdot AS \quad (9)$$

where:

$$P = \begin{bmatrix} D_1 & D_2 & D_3 & D_4 & D_5 & 0 & A_2 & A_3 & A_4 & A_5 & A_6 \\ E_1 & E_2 & E_3 & E_4 & E_5 & 0 & B_2 & B_3 & B_4 & B_5 & B_6 \\ 0 & F_2 & F_3 & F_4 & F_5 & 1 & C_2 & C_3 & C_4 & C_5 & C_6 \end{bmatrix} \quad (10)$$

and

$$AS^T = \begin{bmatrix} a_1 & a_2 & a_3 & a_4 & a_5 & s_1 & s_2 & s_3 & s_4 & s_5 & s_6 \end{bmatrix} \qquad (11)$$

For the designed index finger mechanism (see Figure 2), there are some customized values that equal zero, as follows:

- the axial displacements: $s_3 = 0$; $s_4 = 0$; $s_5 = 0$
- the distances between axes versors: $a_1 = 0$; $a_2 = 0$; $a_5 = 0$.

Results of the kinematic analysis are further presented in Section 3. The mechatronic system for the upper limb has been designed so that it enables finger and forearm motion like that of a real limb as much as possible. Each finger is driven by three micromotors, and its mechanism allows for complex motion, in both the horizontal and vertical planes. The hand and forearm are to be driven by two motors each (and, if required, by assigned gearboxes), and their motion will be possible in two planes (horizontal and vertical).

From the point of view of the system's appearance and its components' dimensions, it has been envisaged to look similar to a real hand. The reverse engineering technique has been used to scan a real hand and to obtain its surface dimensions. The technique used was laser scanning, the specific equipment was a MetraSCAN 3D scanner, and the scanning process did not last more than 3 min. Still, if any slight movement of the real hand occurs while scanning, the acquired data (point cloud) generate a contour surface with errors. This is why the process had to be repeated few times till a correct and complete surface had been generated.

The scanning process was performed using a 0.1 mm mesh resolution at a measurement rate of 800,000 measurements/s. The finished scanned body was achieved by aligning 3 scans, with each scan containing overlapping areas. The resulting body can be seen in Figure 3b,c.

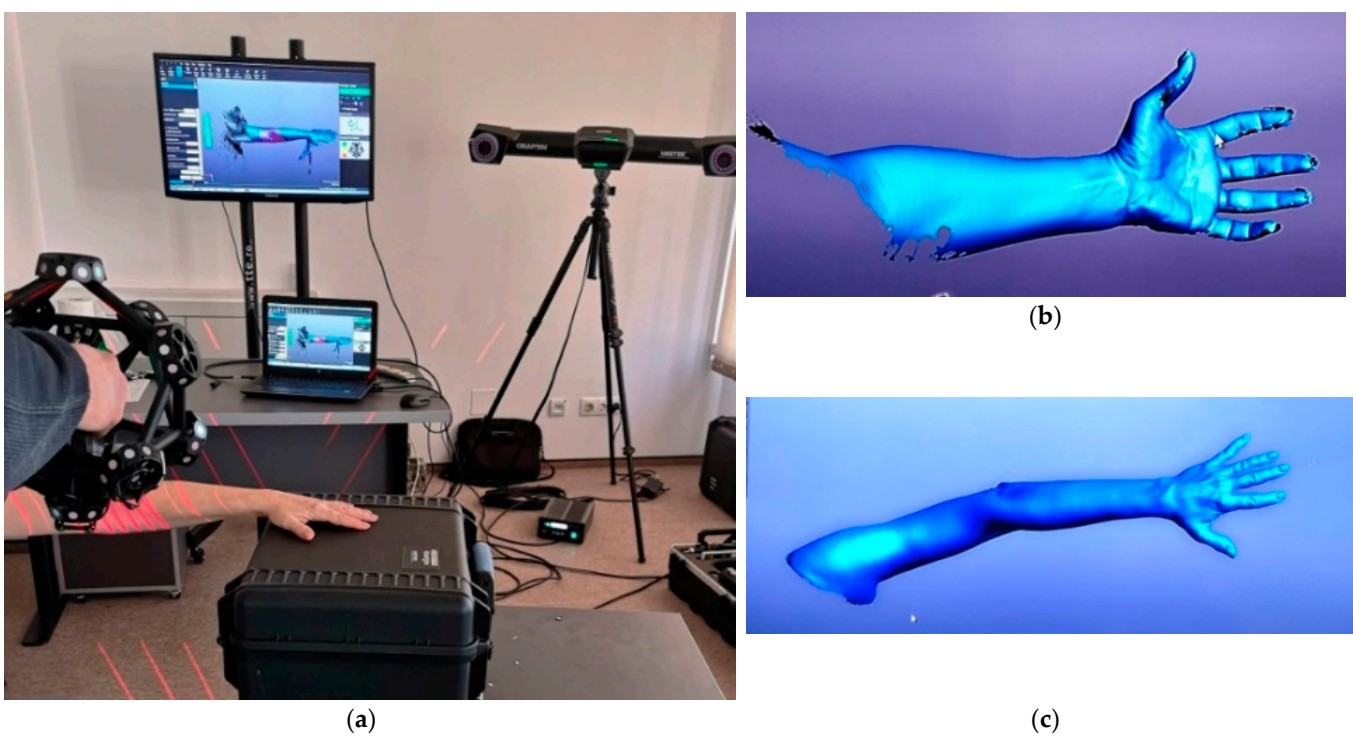

(a)

(b)

(c)

**Figure 3.** Reverse engineering technique for the real upper limb: (**a**) scanning process, (**b**) 3D scan surface—with errors, (**c**) 3D scan surface—accurate.

After each scan, the concerned body and also the table body on which the hand is resting (see Figure 3a) were cleaned by different surplus meshes by using the proprietary

software (VXmodel) and different types of commands and functions within the software. The images taken while scanning are presented in Figure 3.

Once the surface of the real upper limb was obtained (by combining and aligning 3 different scans of the same hand), further processing to determine the hand and fingers' dimensions was performed. Information about the dimensions and contour characteristics (shape and size) of the hand were obtained by using reverse engineering software (Geomagic DesignX and Solidworks) and plane generation, as can be seen in Figure 4. For this process, we used 43 planes from the tips of the fingers to the back of the hand.. The planes were generated using the offset function, and the distance between two consecutive planes was set to 10 mm.

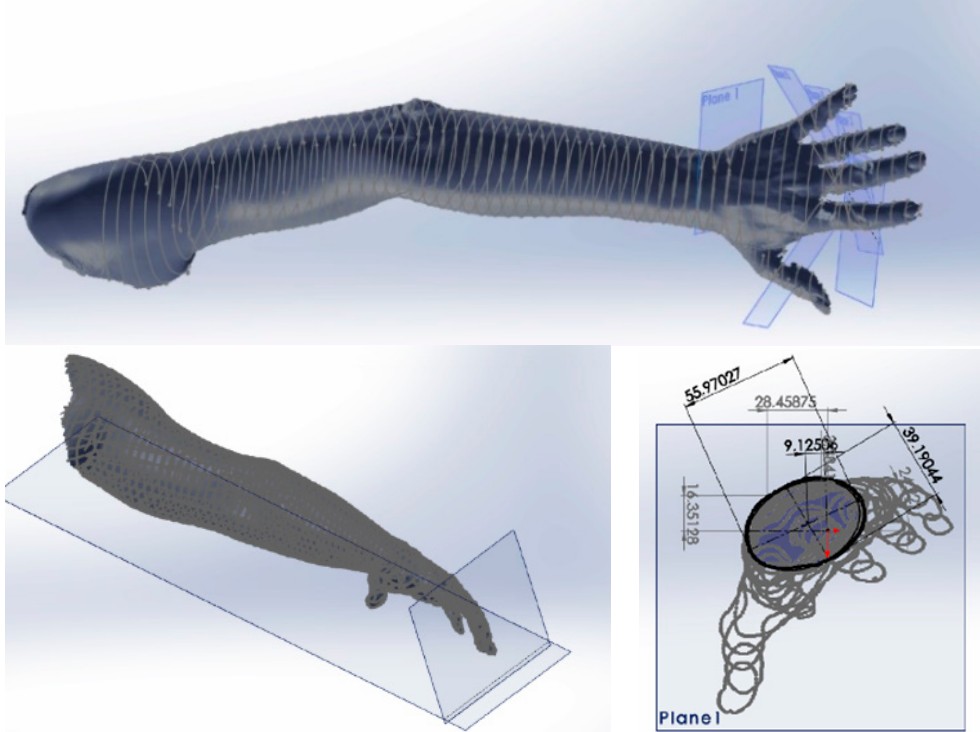

**Figure 4.** Reverse engineered real upper limb surface—transversal plane contour characteristics

If the interest on dimensions referred to the horizontal plane, the scanning process, using the proprietary software (VXmodel), generated a triangle mesh in real time through the alignment or registration method. This process can be seen in Figure 5a. After this is done, the dimensions and dimension parameters were obtained by using reverse engineering software (Geomagic DesignX) and specific tools, such as in Figure 5b. Then, a sketch was generated and parametrized, as in Figure 5c.

The command and control system for the upper limb prosthesis is embedded and designed for customized applied mechatronics application.

Available applications of this prothesis are in the social–medical fields for people with upper limb amputations (hand, forearm or, arm) who would be closer to "normal" everyday life. Another application would be in the field of CBRNE (Chemical, Biological, Radiological, Nuclear, and Explosives) security, for taking samples and/or the manipulation of harmful products in dangerous environments (radiations, chemicals). Still, this presented research is focused on applications for improving the quality of life of people with upper limb disabilities via prosthesis.

This designed system uses data acquisition and control, the data being received from sensors (pressure, tilt, temperature, etc.) and further processed by a micro-controller to manage the micro-motors' control. Depending on the missing part of the human body, the

system's design and structure varies from the point of view of the sensors' type, number, and position; micro-motors; micro-controllers; electric circuit pattern, etc.

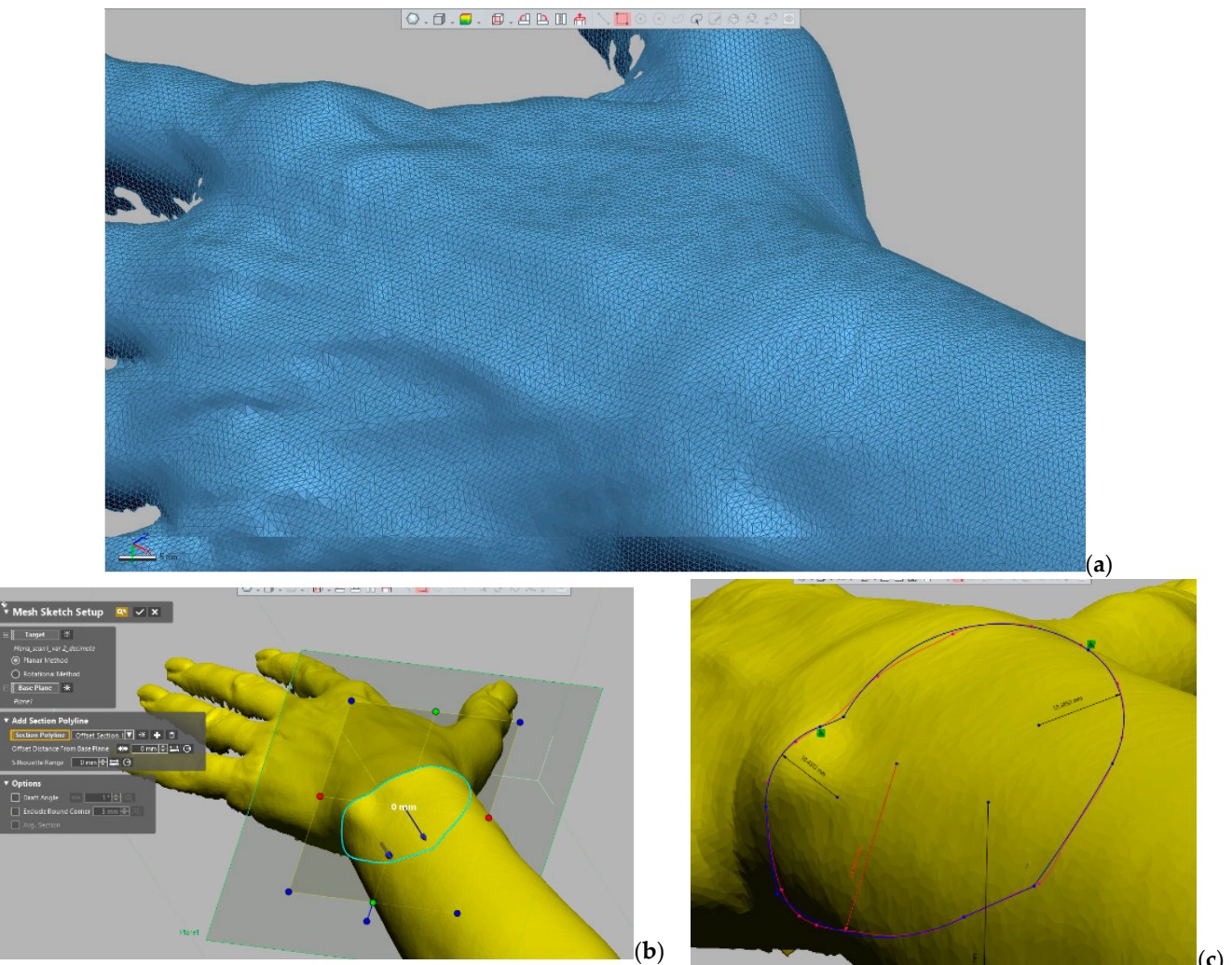

**Figure 5.** Reverse engineered real upper limb surface—horizontal plane contour characteristics: (**a**) triangle mesh, (**b**) dimensions obtained, (**c**) parametrization.

The algorithm scheme of the upper limb prosthesis is shown in Figure 6. The novelty of this concept is that, for each of the five fingers, there are three micro-motors to generate motions (not servomotors, as with most of the available myoelectric prostheses), so that the combination of software and hardware (including the mechatronic system) is specific to this prosthesis. The fingers' control is accurate and complex due to the complexity of motions (in the horizontal and vertical planes).

Basically, the main components of the system are:

- sensors—for peripheral control (pressure sensors for the gripping force of an object, temperature sensors for estimating the temperature of the object to be gripped, and tilt sensors for the inclination angle of the fingers' phalanges);
- drivers for the micro-motors;
- micro-motors (for the fingers' motion) and motors (for the hand and forearm motion) with an assigned demultiplication gearbox, if required;
- microcontroller—a microcomputer with digital and analogic inputs and a GPIO (General-Purpose Input/Output) port.

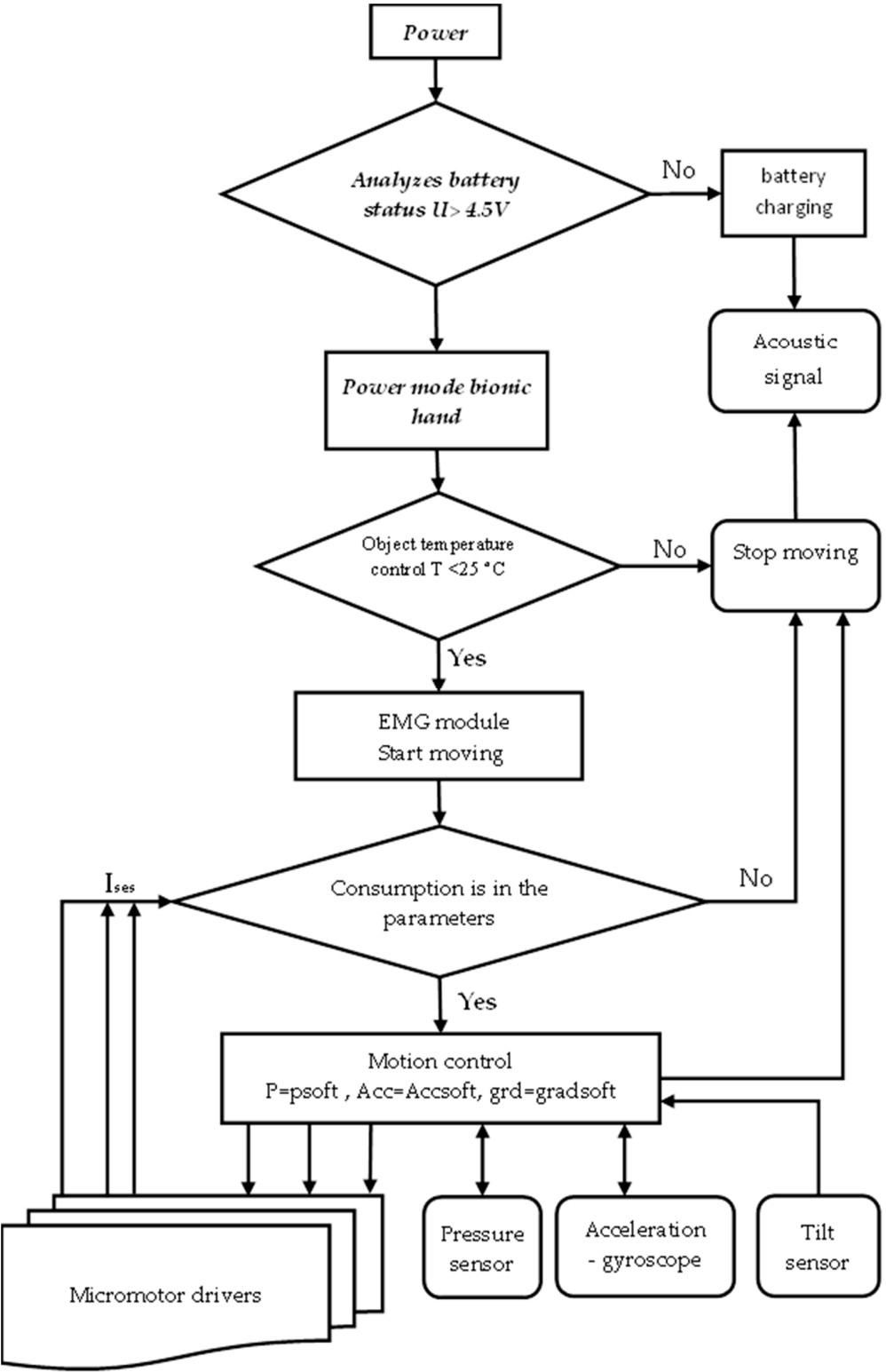

**Figure 6.** The algorithm scheme of the upper limb.

The whole system is conceived as an open hardware platform with high versatility. A sound or vibration warning system could be added if visually impaired people would need the prosthesis.

The main aspects of how the customized prosthesis is designed to work are presented in Figure 7. The first step is that of checking the battery status. If it is necessary, then the battery should be plugged in via the USB port. This port also enables writing to the

flash memory, operational maintenance, adjustment/repairs the system, and updating and improving the prosthesis' system.

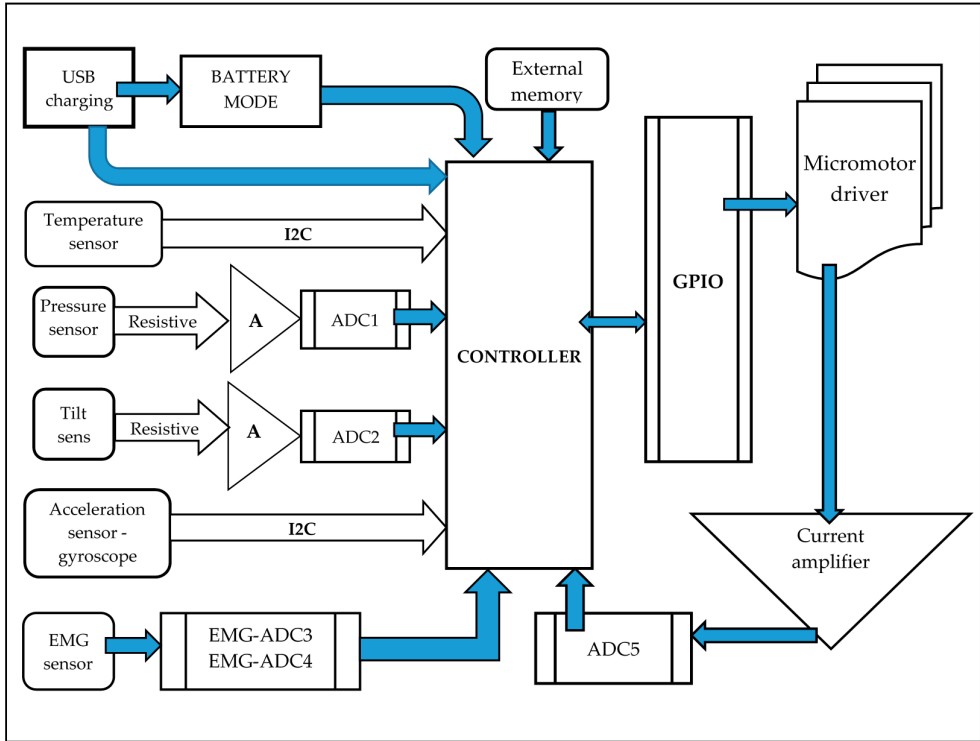

**Figure 7.** The block scheme for the designed prosthesis.

The temperature sensor is an infrared active digital sensor with an I2C communication protocol, and is aimed at determining the temperature of the object going to be gripped. The gyroscope–accelerometer sensors are aimed at determining the angular position of the prosthesis and its motion acceleration. The pressure and tilt sensors are resistive, and for higher conversion accuracy they are connected to an ADC (Analog-to-Digital Converter) by an instrumentation amplifier. If there is not enough digital inputs, then different ADC components will be used (in the case of multi-channel EMG).

The EMG sensor (MyoWare) is aimed at initiating the motion. It could be multi-channel EMG for higher sensitivity.

Micro-motor drivers get commands from the controller via a GPIO interface. The feedback loop is aimed at controlling the energy consumption by measures on each of the micro-motors. The software–hardware tandem ensures motion control by experimenting with the conditions for the correct gripping of objects that are going to be set. By utilizing software limits, p = psoft, Acc = Accsoft, $^\circ$C < XTemp, $\alpha$ < $\alpha$soft, the system continues with the motion or stops it.

In Appendix A, the customized scheme for the EMG sensor (Figure A1) and the customized schemes for the resistive sensors (pressure, tilt) and for the gyroscope sensor (Figure A2) are presented.

## 3. Results

### 3.1. Results of Kinematic and Kinetosatic Analyses of Index Finger Mechanism

The kinematic scheme of the designed prosthesis' fingers is shown in Figure 8 (where $T$, $T_1$, $T_2$, $T_3$, $T_4$ stands for each fingertip point).

The designed mechanism enables two independent rotational motions for each finger and one rotation for each of the phalanges. Simulation of the fingertips' trajectories, as a result of kinematic analyses (with Matlab), is evidenced in Figure 9.

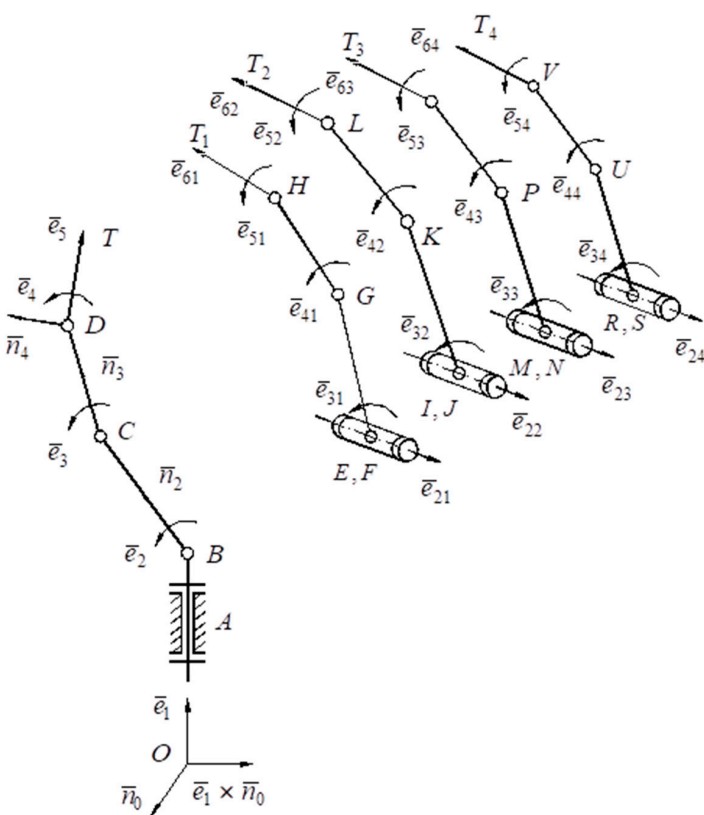

**Figure 8.** Kinematic scheme of the hand's fingers' mechanisms.

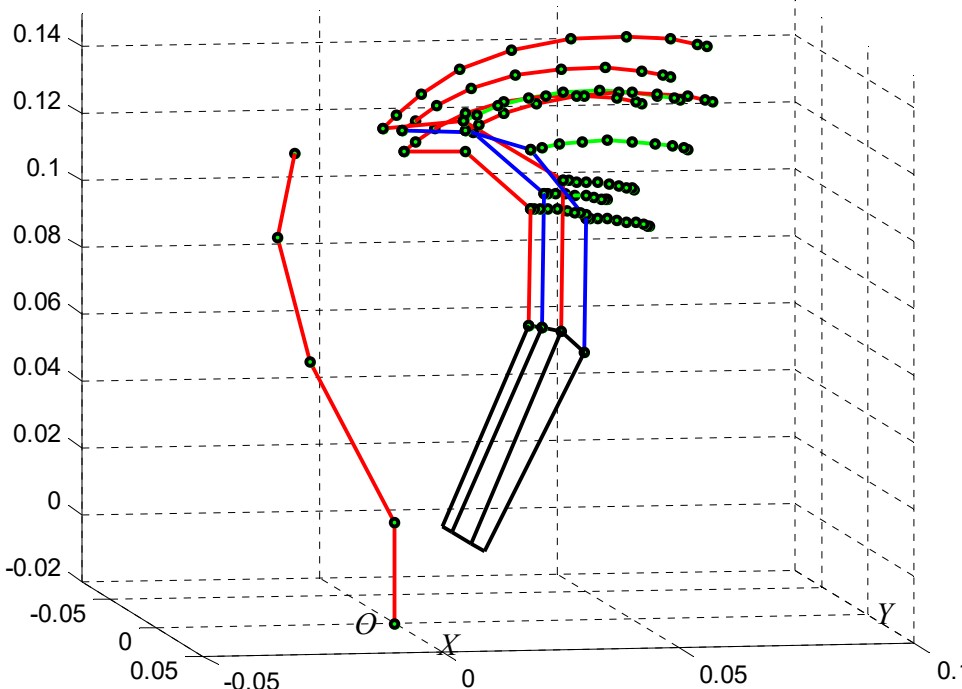

**Figure 9.** Simulation of fingertip trajectories.

The kinetostatic analysis for the customized finger mechanisms was performed. For example, for the index finger mechanism (see Figure 10), there are three phalanges (P1, P2, P3) and the mechanism that enables the rotational motion of the tip and middle phalanx (P mechanism).

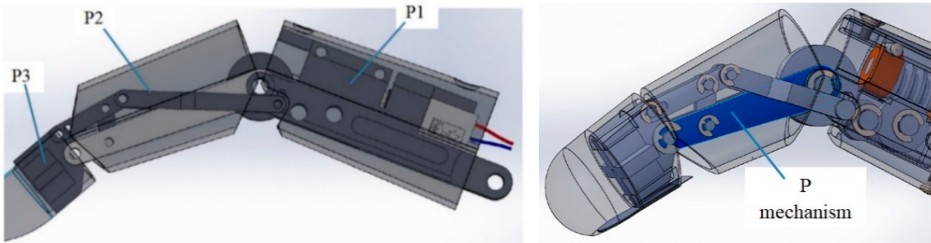

**Figure 10.** Index finger mechanism.

The mechanism is of triad structure, with B, E, and G being the input joints, and C, D, and F being the interior joints (Figure 11). For the index finger model, the kinetostatic calculation was done with Matlab software, assuming that each of the loading forces, $Q_i$ ($i = 1, 2, 3$), are about 20 N (estimated value based on experiments with grasping relatively small objects). The simulation results are presented in Figure 12.

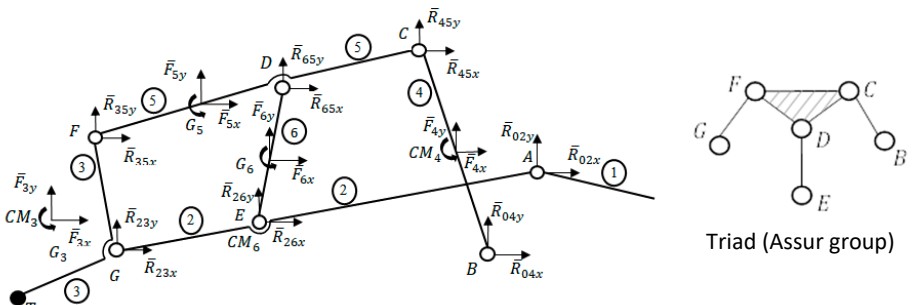

**Figure 11.** Kinetostatic scheme of the index finger mechanism.

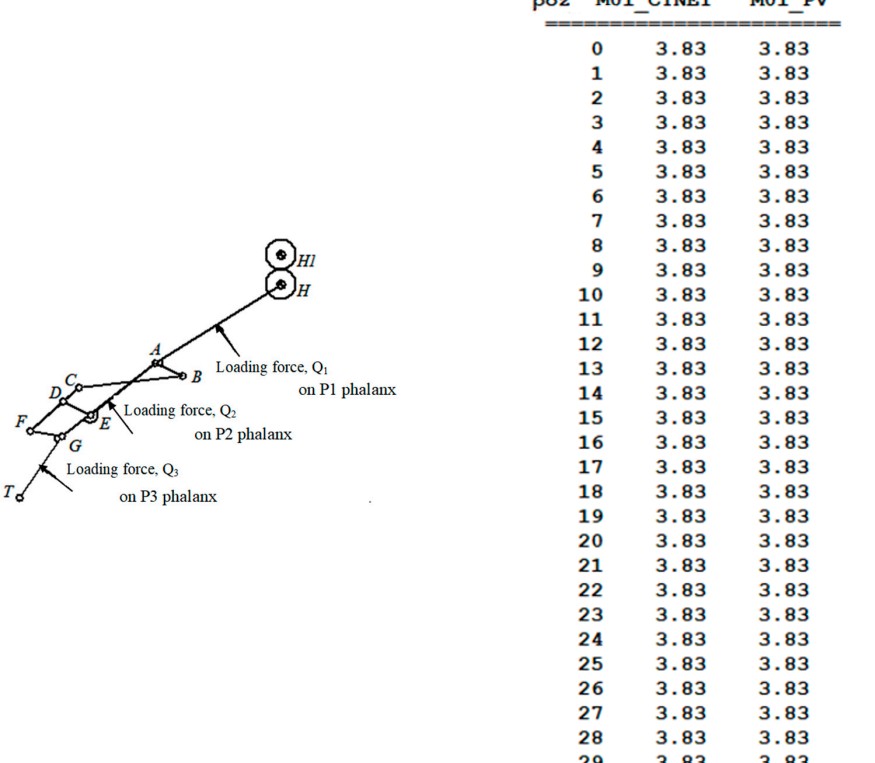

| poz | M01_CINET | M01_PV |
|---|---|---|
| 0 | 3.83 | 3.83 |
| 1 | 3.83 | 3.83 |
| 2 | 3.83 | 3.83 |
| 3 | 3.83 | 3.83 |
| 4 | 3.83 | 3.83 |
| 5 | 3.83 | 3.83 |
| 6 | 3.83 | 3.83 |
| 7 | 3.83 | 3.83 |
| 8 | 3.83 | 3.83 |
| 9 | 3.83 | 3.83 |
| 10 | 3.83 | 3.83 |
| 11 | 3.83 | 3.83 |
| 12 | 3.83 | 3.83 |
| 13 | 3.83 | 3.83 |
| 14 | 3.83 | 3.83 |
| 15 | 3.83 | 3.83 |
| 16 | 3.83 | 3.83 |
| 17 | 3.83 | 3.83 |
| 18 | 3.83 | 3.83 |
| 19 | 3.83 | 3.83 |
| 20 | 3.83 | 3.83 |
| 21 | 3.83 | 3.83 |
| 22 | 3.83 | 3.83 |
| 23 | 3.83 | 3.83 |
| 24 | 3.83 | 3.83 |
| 25 | 3.83 | 3.83 |
| 26 | 3.83 | 3.83 |
| 27 | 3.83 | 3.83 |
| 28 | 3.83 | 3.83 |
| 29 | 3.83 | 3.83 |

**Figure 12.** Kinetostatic analysis of the index finger—final results for active torque.

There were two methods (kinetostatic and virtual power) used for determining the driving torque values (in N·m), both resulting in identical values. This proves the correct calculation results.

### 3.2. Results of Design and Model of the Mechatronic System

Functionally, the design of the mechatronic system was focused on the similarity with real (people's) upper limb motions ability. Each of the hand's fingers performs two rotational motions for the three phalanges of the whole finger, while the tip (P3) and previous phalanges (P2) (see Figure 10) perform one rotation. The hand and the forearm can each perform two independent rotational motions, one similar to the wrist and the other similar to the elbow.

The motion mechanisms are complex and innovative from the point of view of motion transmission components, particularly due to the small dimensions and high geometric precision (tolerances of 0.01 mm, at least). This final aspect (small dimensions) is because of the limited available space for the components, considering the contour characteristics determined by the reverse engineering technique.

Some consequent stages and types of the mechatronic system design are presented in Figure 13.

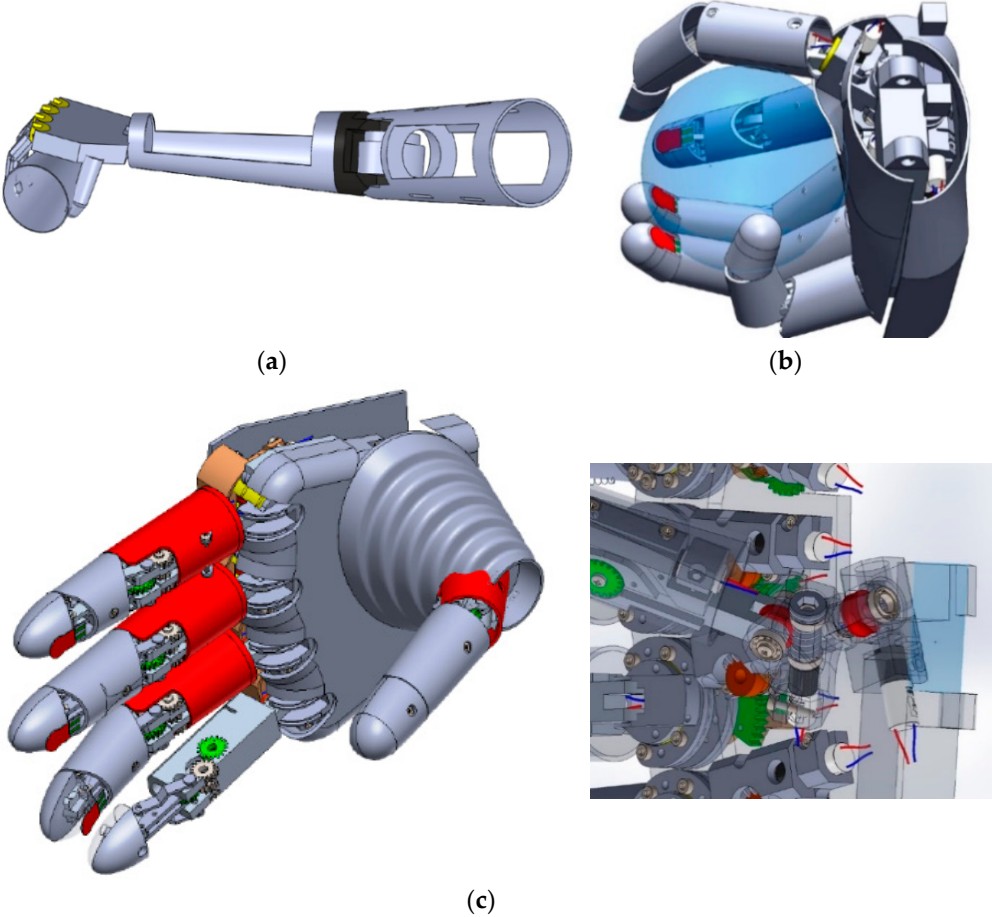

**Figure 13.** Mechatronic system design: (**a**) first design type, (**b**) intermediate design type, (**c**) up to date design—motion transmission components

### 3.3. Results of Command and Control System

The command and control system is a complex one, briefly designed according to its block diagram presented in Figure 14. Basically, the EMG sensor [25] mounted on the existing part of the upper limb will generate the signal for motion toward and/or grabbing

of an object (cup, ball, pencil, etc.). The resistive and infrared sensors [26] will acquire and send to the controller the data regarding prosthesis position, acceleration, pressure on the gripped object, temperature of the object, and motion towards the object. The motion toward the object is then available via micro-motors acting on mechanical structure (gears, joints, etc.) and assisted by the gyro sensors. Once the target is reached according to information on the software set values (psoft, Accsoft, °XTemp, αsoft) and real values from the sensors, the prosthesis motion will stop or reverse.

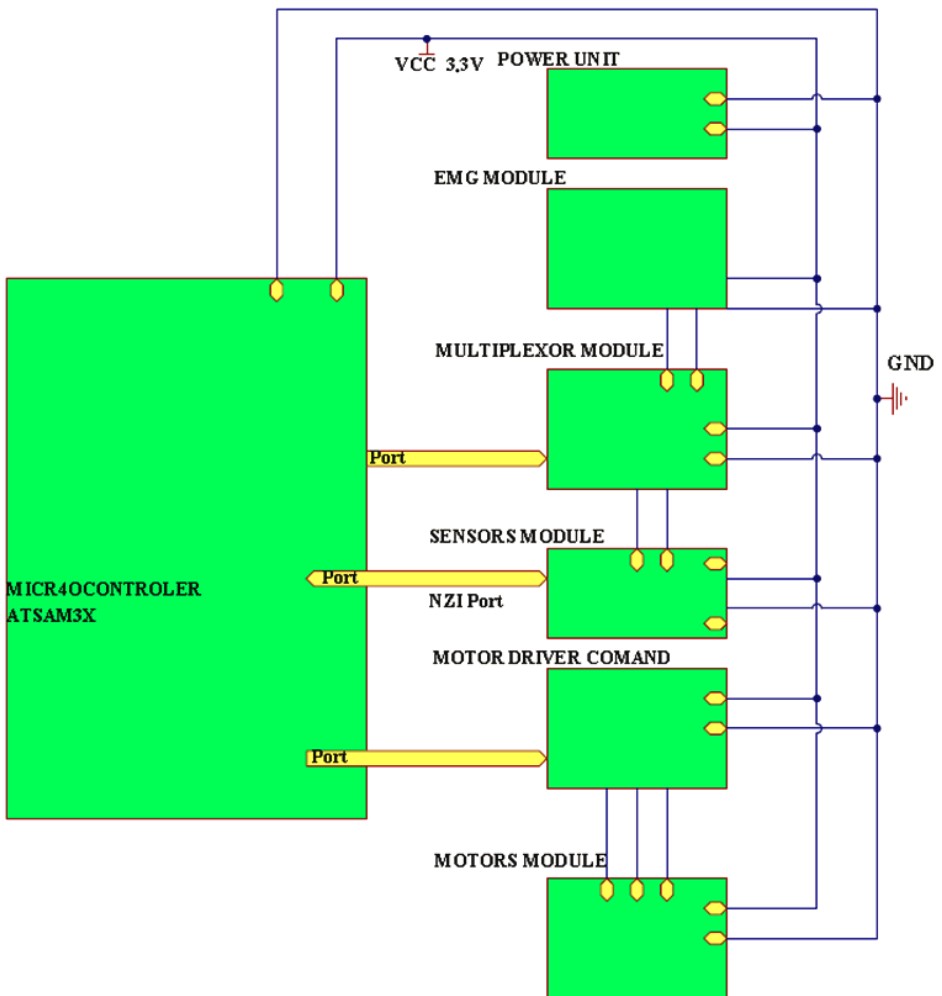

**Figure 14.** Block diagram of the command and control system for the upper limb prosthesis.

This paper presents research results focused on the concept, design, prototyping and initial tests of the prosthesis. The command and control system's testing refers to the preliminary testing of the sensors and micro-motors on a small platform, such as Arduino.

One conclusion when testing the resistive sensors pointed out the need for careful selection of the base material, that of the fingers' surface. This is because it should not be extremely rigid—so as to allow for the correct fixture of flexible sensors, and not too soft—so that the material does not to overtake deformation and thus influence the resistive sensor's specific deformation. Images taken on the preliminary experiments are shown in Figure 15. In Appendix B the program customization for the display of conventional units for force [N] (Figure A3) and tilt angle [°] (Figure A4) is presented.

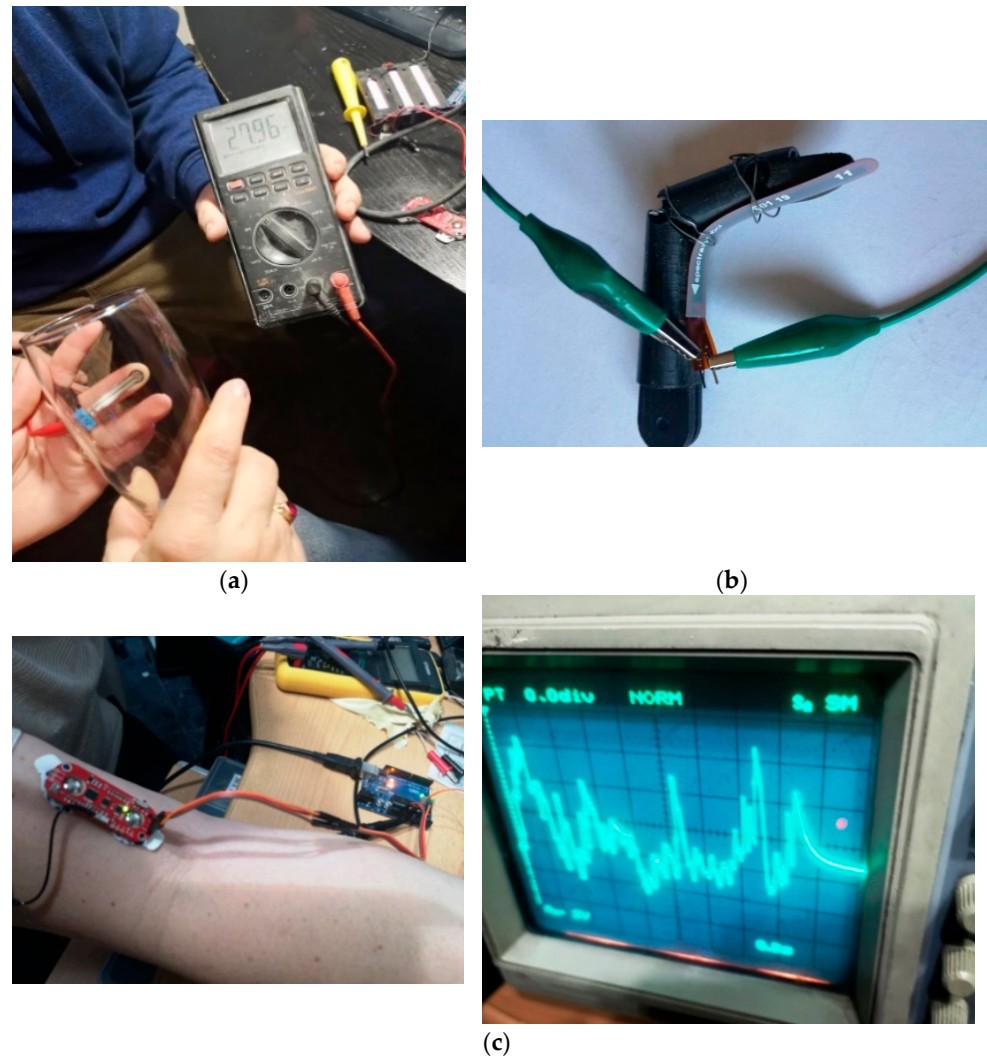

**Figure 15.** Experiments on the sensors. (**a**) FSR pressure sensors—for force, (**b**) tilt sensor—for finger inclination angle, (**c**) EMG sensor test.

### 3.4. Prototyping the Mechatronic System Parts

One of the modern, and relatively cost affordable, techniques for prototyping is 3D printing. Mechanical components of the mechatronic system were prototyped using several rapid prototyping techniques due to various offers on the market. The scope was to obtain accurate prototypes for the parts made of fit materials (polymers, resins) and, not least, with moderate costs.

The first technique used was FDM (fused deposition modeling), the material was biodegradable thermoplastic polymer, PLA, and the equipment was Creality Ender 3-Pro [27]. Only the exterior surface of the index finger was printed for preliminary tests (see Figure 16). The prototype proved to be accurate enough for manually testing finger mobility and for preliminary tests of the pressure sensors. Still, the technique had limitations because of shape deviation from the nominal values (outside the acceptable upper/lower limits) and the material's characteristics (relatively high density and, consequently, not lightweight enough).

One more step in prototyping the mechanical components was that of considering only the motion of thumb and index fingers—due to their importance in handling everyday objects. If the mechatronic structure with only these two fingers "worked" correctly, then it could be assumed that all the rest of the structure would do so.

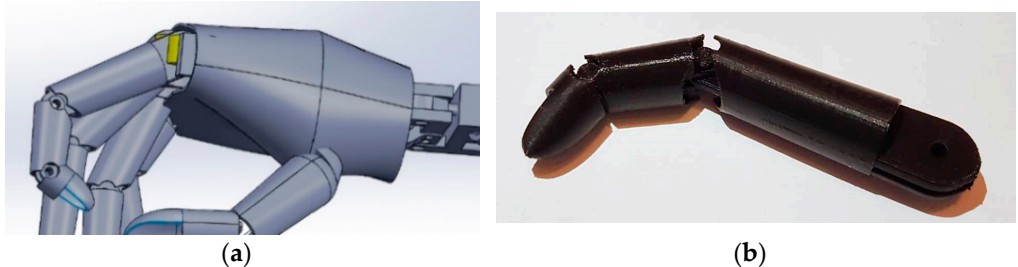

**Figure 16.** FDM prototype of the index finger surface. (**a**) 3D model, (**b**) FDM prototype.

The 3D model to be prototyped, conventionally named, 2-FPRINT, is presented in Figure 17.

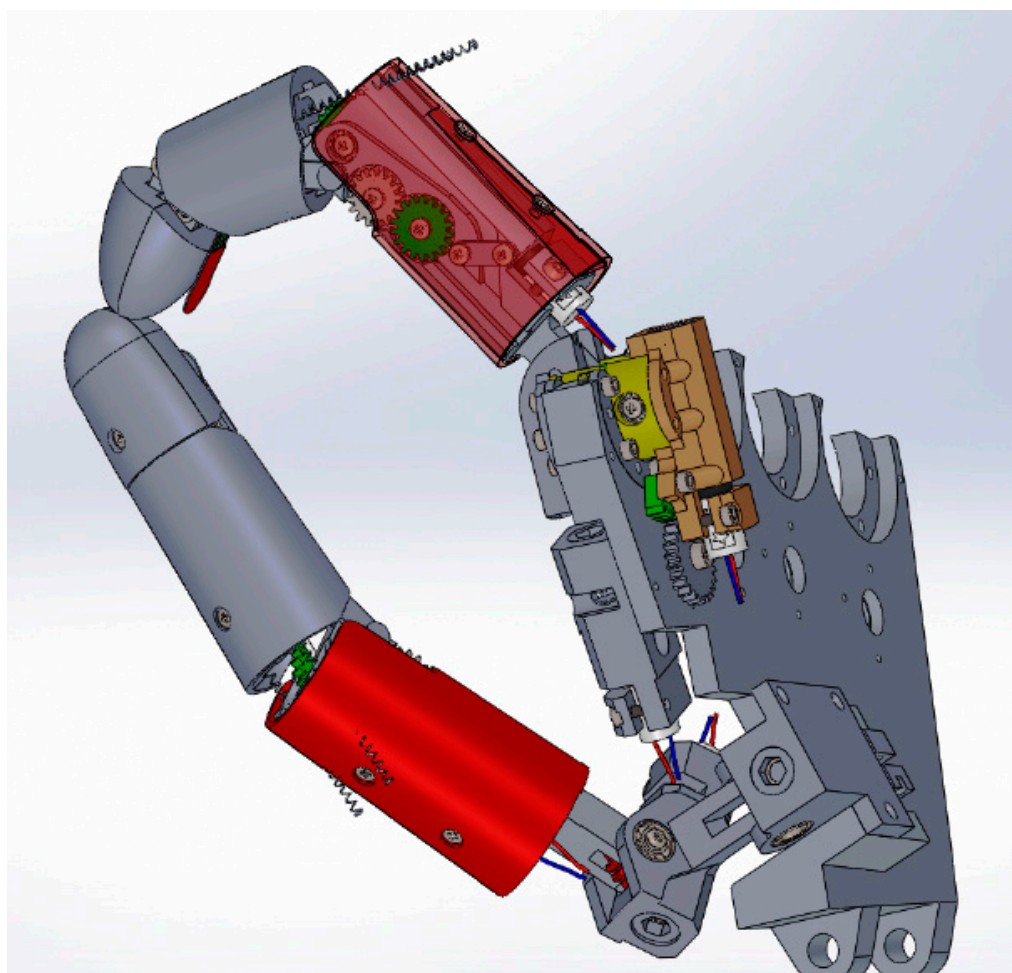

**Figure 17.** 3D model of the two fingers prototype, 2-FPRINT.

The first prototyping technique considered was the one from EnvisionTEC that is based on "3SP (Scan, Spin and Selectively Photocure) technology to quickly 3D print highly accurate parts from STL files, regardless of geometric complexity" [28]. The printing equipment is Envisiontec ULTRA 3SP, and the resin is a special one, EnvisionTEC's premium E-Model Light [29], used mainly for orthodontic purposes. Some of the complex components were printed, so as to test this 3SP technique's results. Images taken while prototyping are shown in Figure 18.

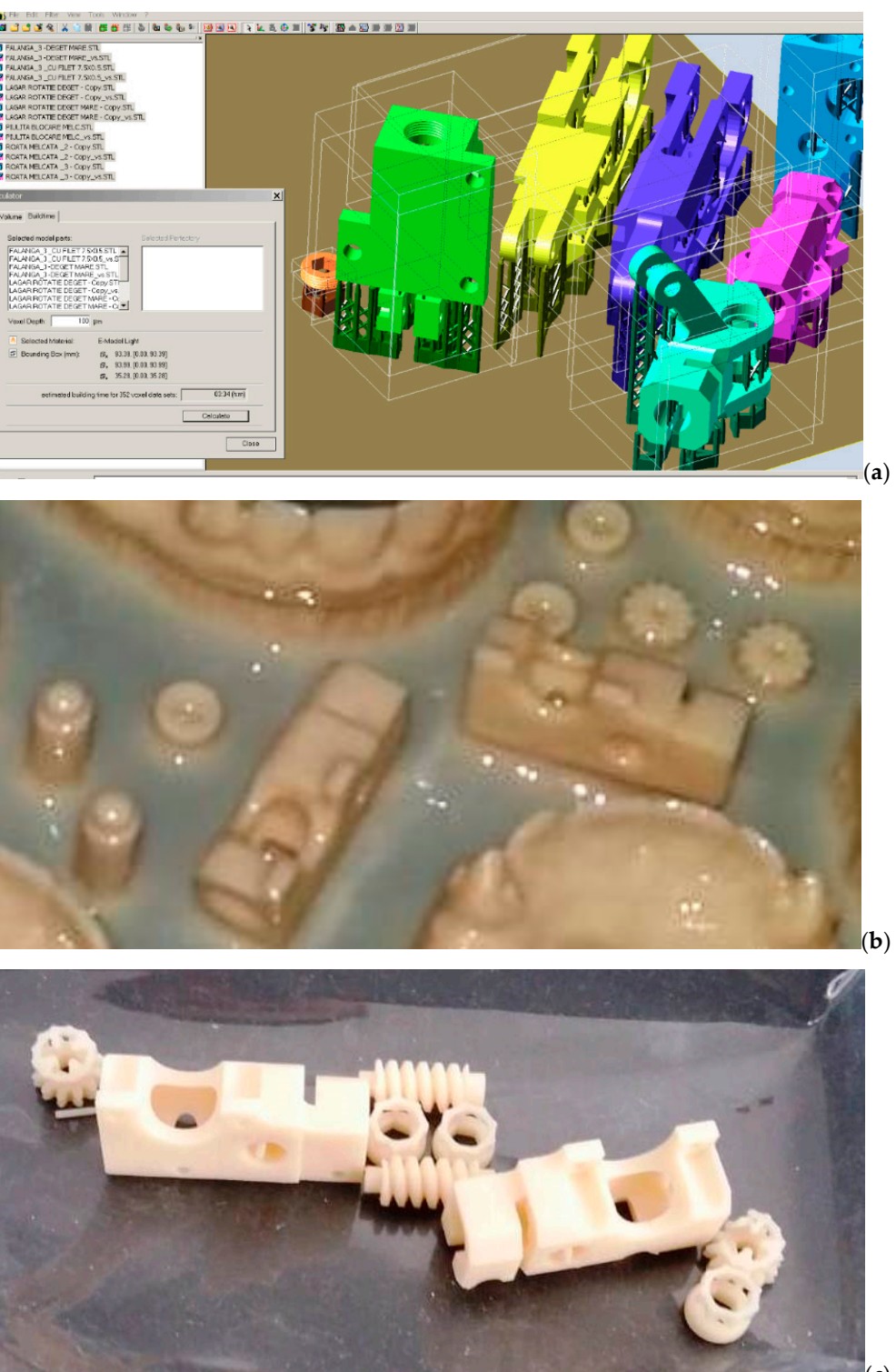

**Figure 18.** 3SP printing of mechanical parts. (**a**) Model construction and position, (**b**) models at the final printing process, (**c**) printed parts.

After having the printed parts photocured, one important task was that to check their geometric precision (see Figure 19). Dimensions of the holes for mounting the rolling bearings and the micro-motors, respectively, as well as their tolerances, were determined. Additionally, the thread dimensions (pitch of 0.5 mm) were checked. Furthermore, the worm–worm gear transmission system was tested.

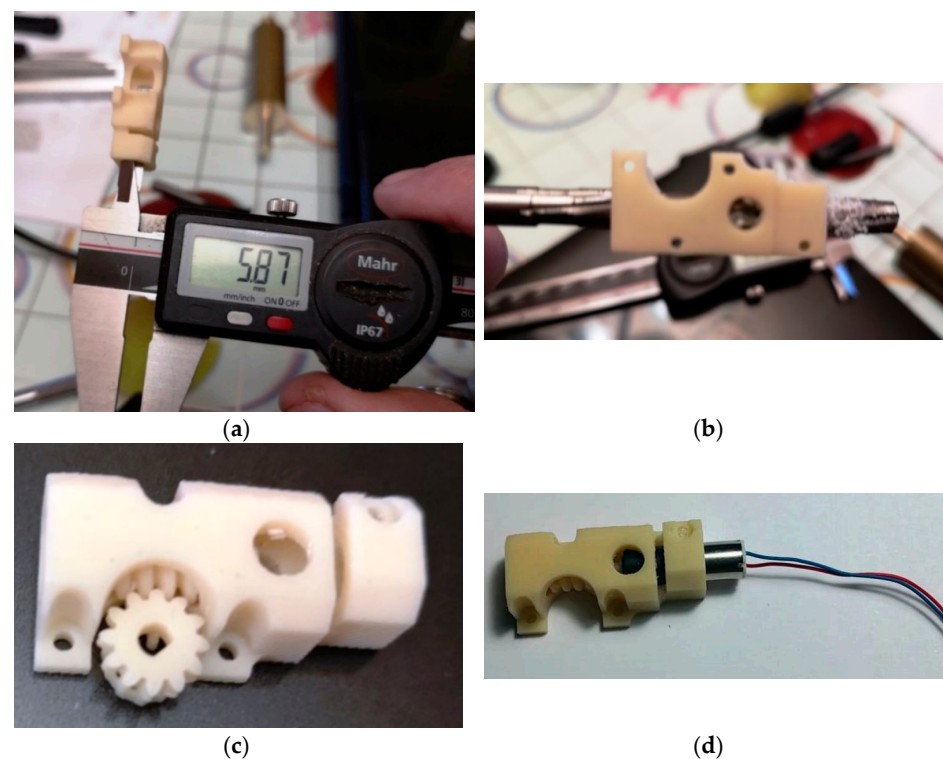

**Figure 19.** Checking the geometric precision of the 3SP printed parts. (**a**) Whole dimension in a section: 5.87, instead of 6 mm, (**b**) manual reamer to adjust the whole, (**c**) the gear–worm gear needed to be further adjusted for correct motion transmission, (**d**) incorrect assembly of the micro-motor and the case part.

The conclusion was that this printing technique did not prove to be adequate for the accurate and small parts of the mechatronic system designed. There were tolerances larger than the designed ones (0.02 mm), and most hole dimensions were smaller than nominal, i.e., smaller than the lower limits. Moreover, all exterior dimensions resulted in higher than upper limit values, so that additional processing would be required. The thread pitch was not correct, and almost proved to be threadless. It could not be assigned a set of "rules" for overcoming these geometric problems, as the deviations from nominal dimensions were not similar for all the printed parts.

Another printing technique used for the rapid prototyping of the mechatronic system parts was DLP (digital light processing). DLP technology uses a digital projector to project an image of the layers, processed with specialized software, across the entire build platform, curing the whole surface of the layer through a single flash.

The printed components of the mechatronic system were built using the printer "Figure 4 Standalone" from 3DSystems. The material used was TOUGH-GRY 10, which is a strong production plastic with printing speeds of up to 10 cm/hour that is extremely stable at high humidity [30].

The mechatronic system's components are very accurate, with each component having multiple fine geometries, and the details needed by the printed parts lead to the usage of a layer resolution of 10 microns.

The materials used by the Figure 4 printer are all resin materials, and, because of this, the parts will need support as the build platform rises from the tank, layer by layer. All the support processing, part slicing, and optimization is performed with the help of 3D Sprint, a slicing platform software used for multiple printers, and printing technology from 3D Systems. The support structure is generated and optimized based on the part position in regard to the XY plane. Then, the downface of the components is calculated based on the premise of angle and slope, as can be seen in Figure 20. Basically, the downface is created based on the default angle of 45° and the gradient of 1° (Figure 20a). Due to the 10 micron

layer height, we could change the angle to 35° and the gradient to 2°. The result of the change can be seen in Figure 20b, and represents the best solution for the parts in regard to the amount of supports needed for printing.

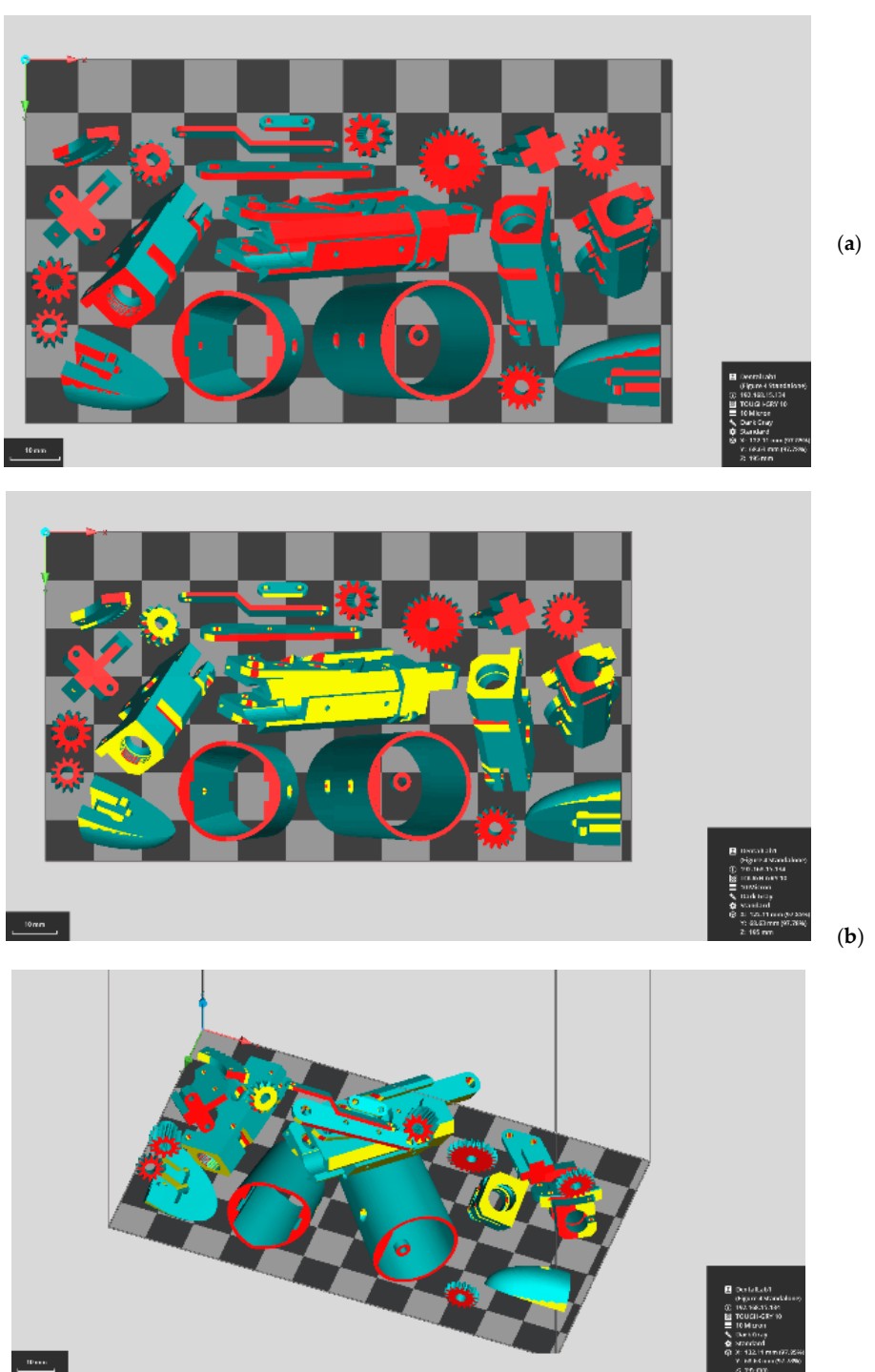

**Figure 20.** The downface verification based on angle and slope. (**a**) Downface of 45°, (**b**) downface of 35°.

Lastly, the support points, pillars, pillar bridges, and base truss are generated. The tip that is joined with the component surface is generated on the premise of penetration length, contact point width, etc., settings that are manually optimized based on the component geometries and utility. The support structure for the printed parts is shown in Figure 21.

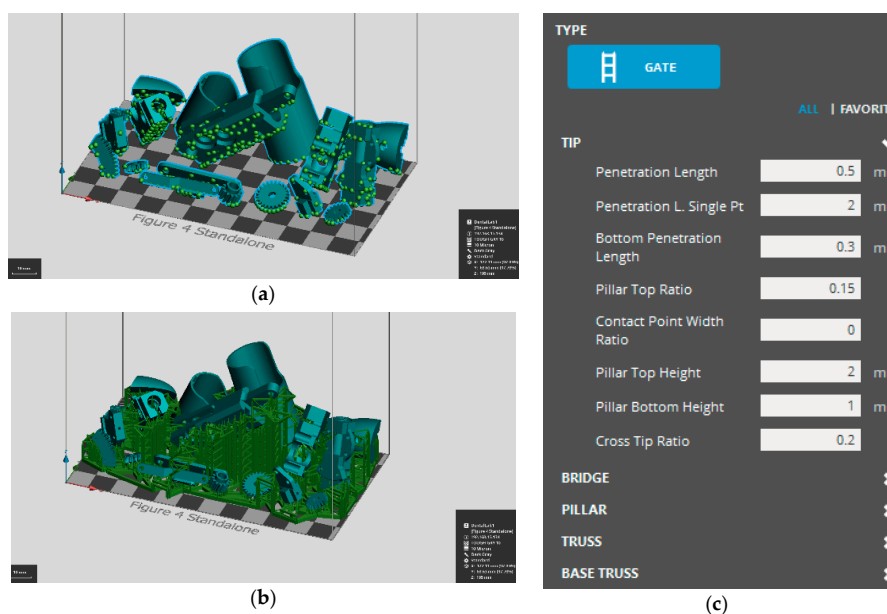

**Figure 21.** Optimization and creation of the support structure. (**a**) Support points generated by the software, (**b**) support pillars based on the generated points, (**c**) support settings.

Once the optimization and virtual build procedure ended, the printing process started. The result of the 3D printing can be seen in the images below (Figure 22).

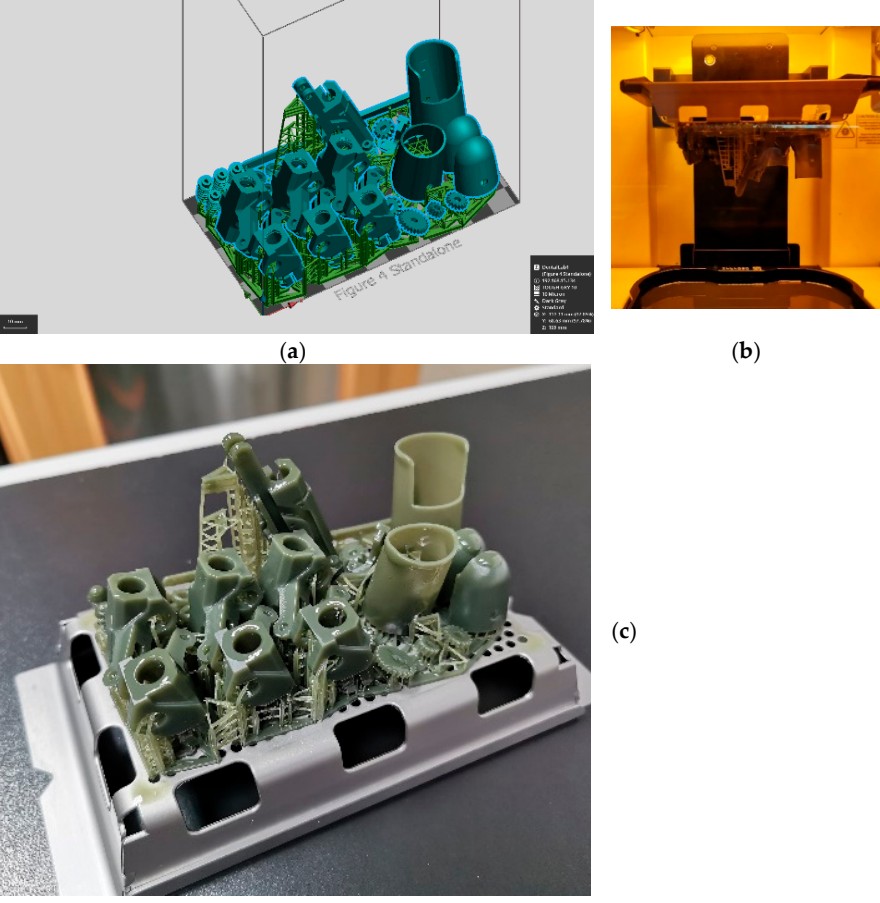

**Figure 22.** Print job finished for mechanical components. (**a**) Build table optimized for 3D print, (**b**) print job finished, (**c**) build platform before alcohol part wash.

The printed components shown in the figure above are what, in 3D printing, is called a "green part". A green part, beside the resin, contains polymers, monomers, and oligomers. The green part, before receiving the mechanical proprieties of the material used in 3D printing, needs to be cured with ultraviolet light in a special oven, as can be seen in Figure 23.

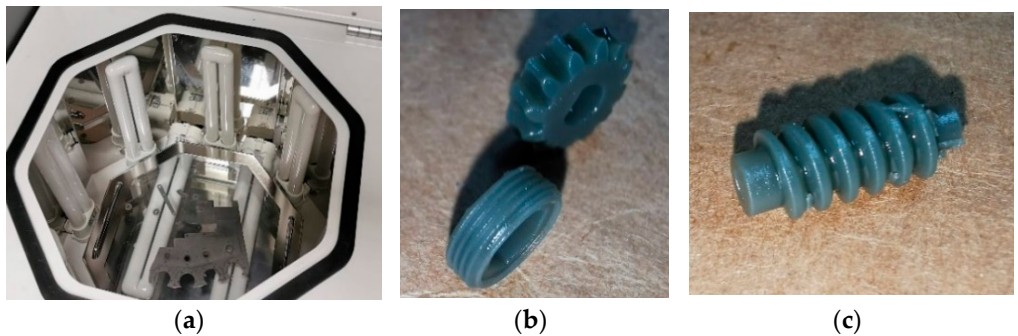

|  (a)  |  (b)  |  (c)  |

**Figure 23.** Printed green part curing and level of detail. (**a**) Curing ultraviolet oven, (**b**) gear and fillet level of detail (thread pitch of 0.5 mm), (**c**) worm gear level of detail (module of 0.6 mm).

After the curing stage, the final parts could be checked for geometric precision. The accuracy of the print dimension was evaluated by comparing the print parts' measurement results to the measurements done in the 3D Sprint software (the same with the designed dimensions in CAD software). Not all components were measured, as for the verification we need only 2–3 parts from each print table. An example of the measurement results is presented in Figure 24.

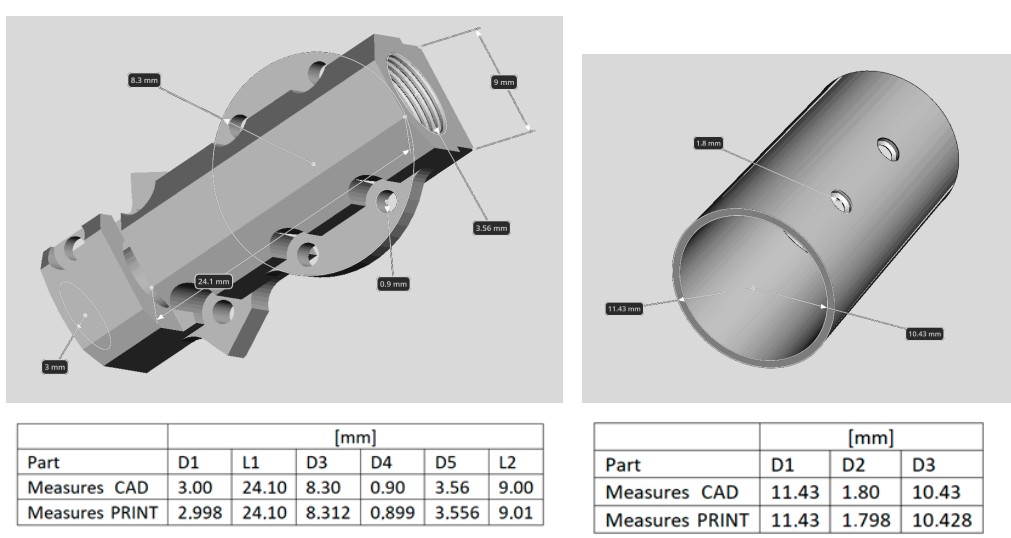

| Part |  | [mm] |  |  |  |  |
|------|-----|-------|------|------|------|------|
|  | D1 | L1 | D3 | D4 | D5 | L2 |
| Measures CAD | 3.00 | 24.10 | 8.30 | 0.90 | 3.56 | 9.00 |
| Measures PRINT | 2.998 | 24.10 | 8.312 | 0.899 | 3.556 | 9.01 |

| Part |  | [mm] |  |
|------|-----|------|------|
|  | D1 | D2 | D3 |
| Measures CAD | 11.43 | 1.80 | 10.43 |
| Measures PRINT | 11.43 | 1.798 | 10.428 |

**Figure 24.** Statistics of the printed parts' dimensions.

The most important and good aspect was that, on the whole, tolerances were small enough (according to lower and, respectively, upper prescribed values) that the plug gauge check was successful from the first test. Assembling the parts (gears, bearings) to test (mechanically, for this moment) the motion transmission proved to function as intended in the prosthesis design (see Figure 25).

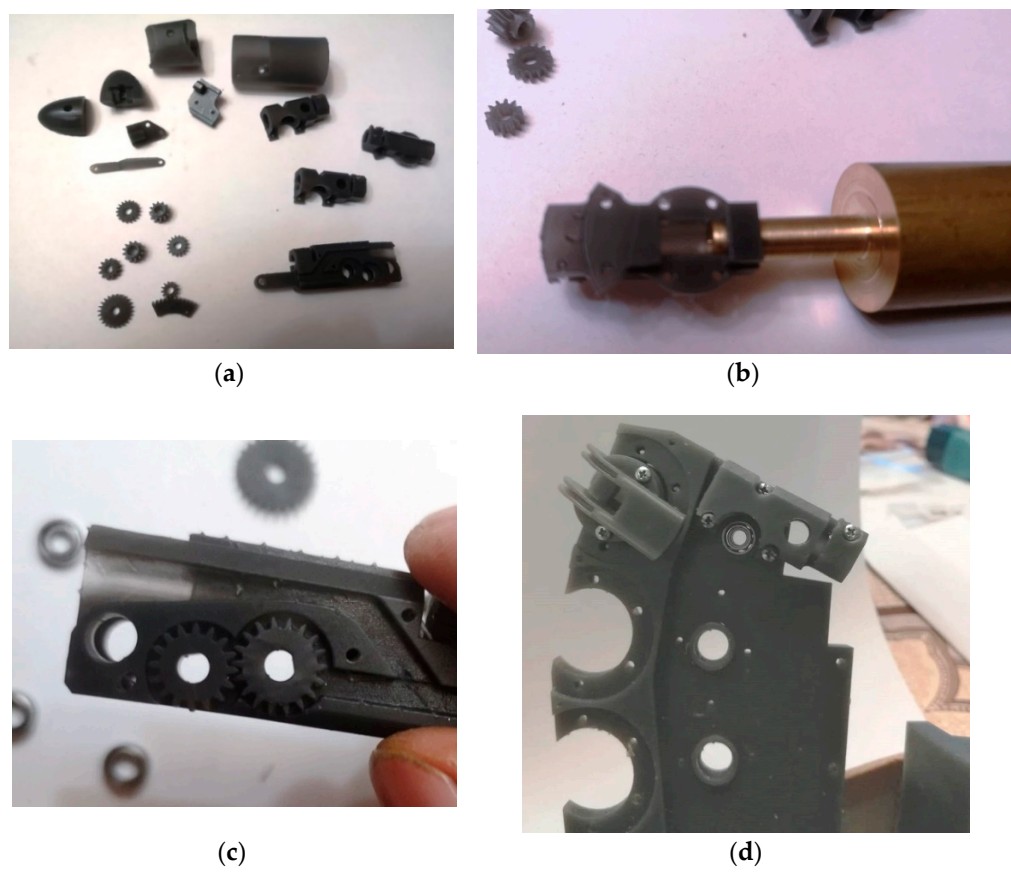

(a)

(b)

(c)

(d)

**Figure 25.** Checking the geometric precision of the DLP printed parts. (**a**) Printed parts, (**b**) plug gauge check, (**c**) correct gear assembly for motion transmission, (**d**) bearings mounted.

The assembly of some DLP printed parts into the 2-FPRINT prototype (see Figure 17), and thus the resulting components of this prototype, is shown in Figure 26.

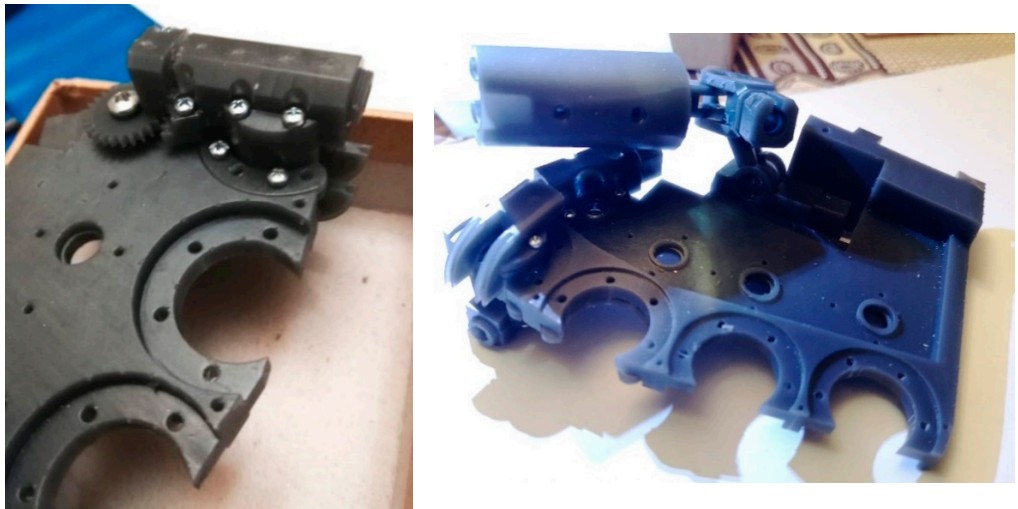

**Figure 26.** Components of the 2-FPRINT prototype.

## 4. Discussion

The research results presented in this paper are focused on the concept and design of a prosthesis for the upper limb.

Customization of this prosthesis is an important issue. Due to the reverse engineering technique, size, dimensions, and other geometric characteristics of a real upper limb would

be achievable with this prosthesis. The 3D scan of the upper limb was performed for one of the authors of this paper. The reverse engineering technique enabled determining the value for any dimension, both in the transversal and horizontal planes. The design of this prosthesis would match, as much as possible, these dimensions, which are the ones of the scanned upper limb. This poses a real challenge for the concept and design of the mechanical components, as there is limited available space when considering the reverse engineered contour characteristics.

The finger mechanism was studied from the point of view of kinematic and kinetostatic analyses. The matrix–vector method, aided by Matlab software, was used to determine the joint and fingertip trajectories. Simulation results for the finger mechanism evidenced correct motions.

The command and control system is embedded and designed for customized applied mechatronics applications. The novelty of this concept is that for each of the five fingers, there are three micro-motors to generate motion (and not servomotors, as in most available myoelectric prostheses). The combination of software and hardware (the mechatronic system included) is specific to this prosthesis. It enables the accurate trajectory of the fingertips and different set rules (gripping pressure, object temperature, acceleration toward the object). The ROM of each joint will be defined once all the tests are completed, and the validation of the results will be performed.

Preliminary testing of the sensors and micro-motors on a small platform, Arduino, was performed for the prototype (two fingers, 2-FPRINT). The conclusion from testing the resistive sensors pointed out the need for the careful selection of the base material, specifically the one for the fingers' surface. The upper limb prosthesis (whole product) will have its PCB, or micro-computer; will be versatile; and will be of small dimensions.

The concept of this upper limb prosthesis is aimed at becoming a high-versatility hardware platform. Optionally, for example, a system for sound or vibration warning could be integrated if a visually impaired person was to need the prosthesis.

Prototyping the mechanical components has been a challenge because of the high accuracy needed for the geometric precision of parts. Several techniques of rapid prototyping have been considered, out of which the DLP (Digital Light Processing) proved to be the right one.

As, for the moment, only obtaining the prototype is intended, there is no ethical approval asked from any authorities. Later in time, after the prototype manufacturing is completed and all tests are correct, validation and, perhaps, IRB approval would be required.

## 5. Conclusions

This paper presents aspects of the concept and design of a customized prosthesis for the upper limb. The software and hardware tandem concept of this mechatronic system enables complex motion (in the horizontal and vertical planes) with accurate trajectory and different set rules (gripping pressure, object temperature, acceleration toward the object).

The concept of this upper limb prosthesis is different from the many existing prostheses existing on the market. The fingers' motion is initiated by EMG sensor signal, and the mechanical components are driven by micro-motors, three for each finger (and not by servo-motors, as with most myoelectric prostheses). The rapid prototyping of mechanical components has been considered instead of classical manufacturing techniques (CNC machining by turning, milling, drilling, etc). There were several 3D printing techniques applied of those available and estimated to fulfil the requirements of parts (high geometric precision and accurate finish). It was only DLP (digital light processing) that enabled the manufacturing of mechanical parts with the conditions prescribed in the design.

The customization of the prosthesis' aspect and dimensions, close to those of a person's real limb, is due to the reverse engineering technique. Further development of this research will aim for the careful selection of materials and manufacturing of the all the mechanical components for the upper limb prosthesis, assembling them into a whole

prototype (including micro-motors, controllers, sensors, etc.), and the complete testing and validation of results.

The upper limb prosthesis prototype will be in accordance with the body features of the person wearing it, will be lightweight, be easy to maintain, and its cost will be a few thousand Euros.

**Author Contributions:** Conceptualization, M.I. and C.R.; methodology, M.M.R. and L.M.U.; software, L.M.U. and L.M.; validation, M.I. and C.R.; formal analysis, M.I. and L.M.U.; investigation, C.R. and M.M.R.; resources, L.M. and C.R.; writing—original draft preparation, M.I. and M.M.R.; writing—review and editing, M.I., M.M.R. and C.R.; supervision, M.I. All authors have read and agreed to the published version of the manuscript.

**Funding:** This research received no external funding.

**Institutional Review Board Statement:** Not applicable.

**Informed Consent Statement:** Informed consent was obtained from all subjects involved in the study.

**Data Availability Statement:** Not reported yet.

**Acknowledgments:** We especially thank CADWORKS International SRL for the support and help given in the elaboration of this work. https://www.cadworks.ro/*** (accessed at 5 January 2021). We also thank the Romanian Academy for their support of this research.

**Conflicts of Interest:** The authors declare no conflict of interest.

**Appendix A**

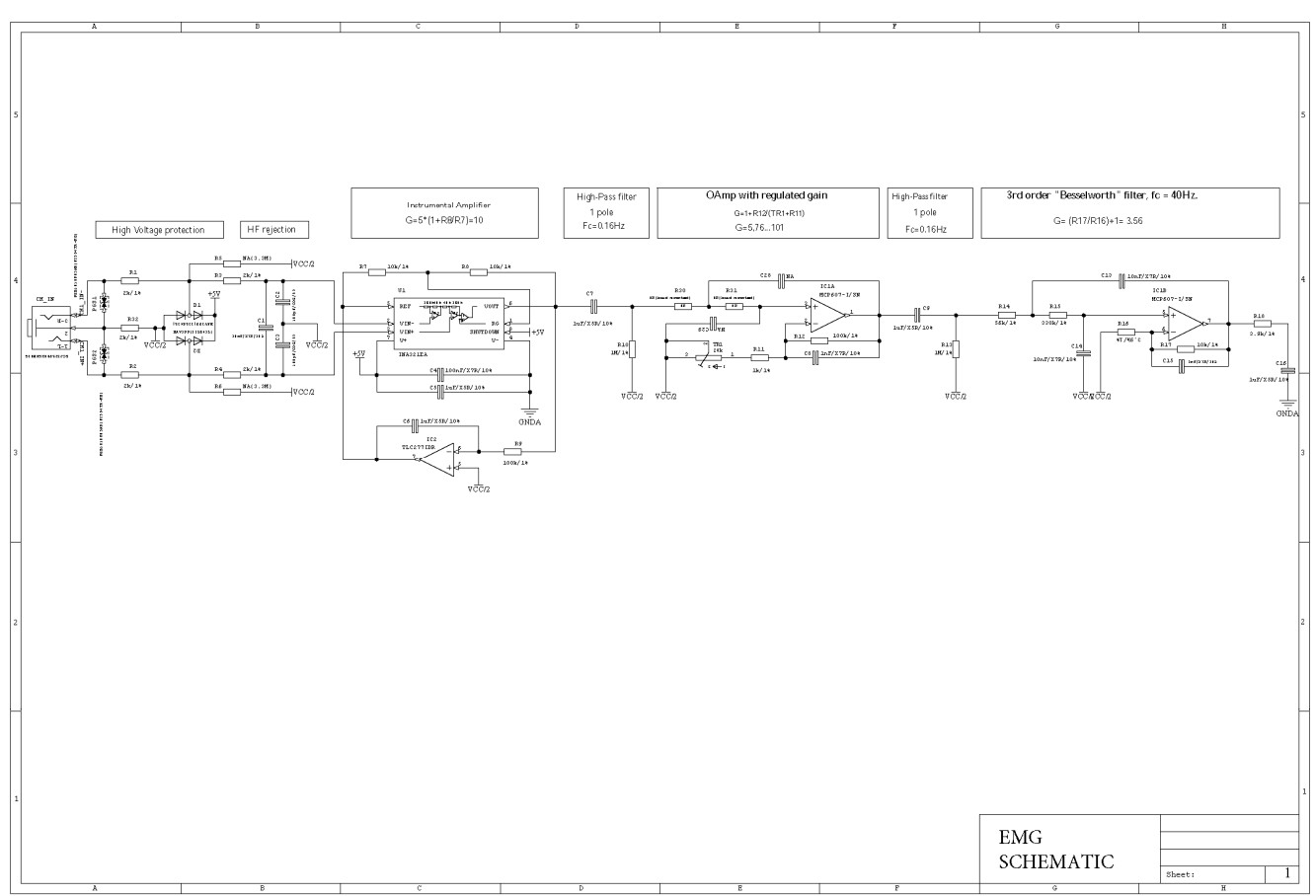

**Figure A1.** EMG schematic.

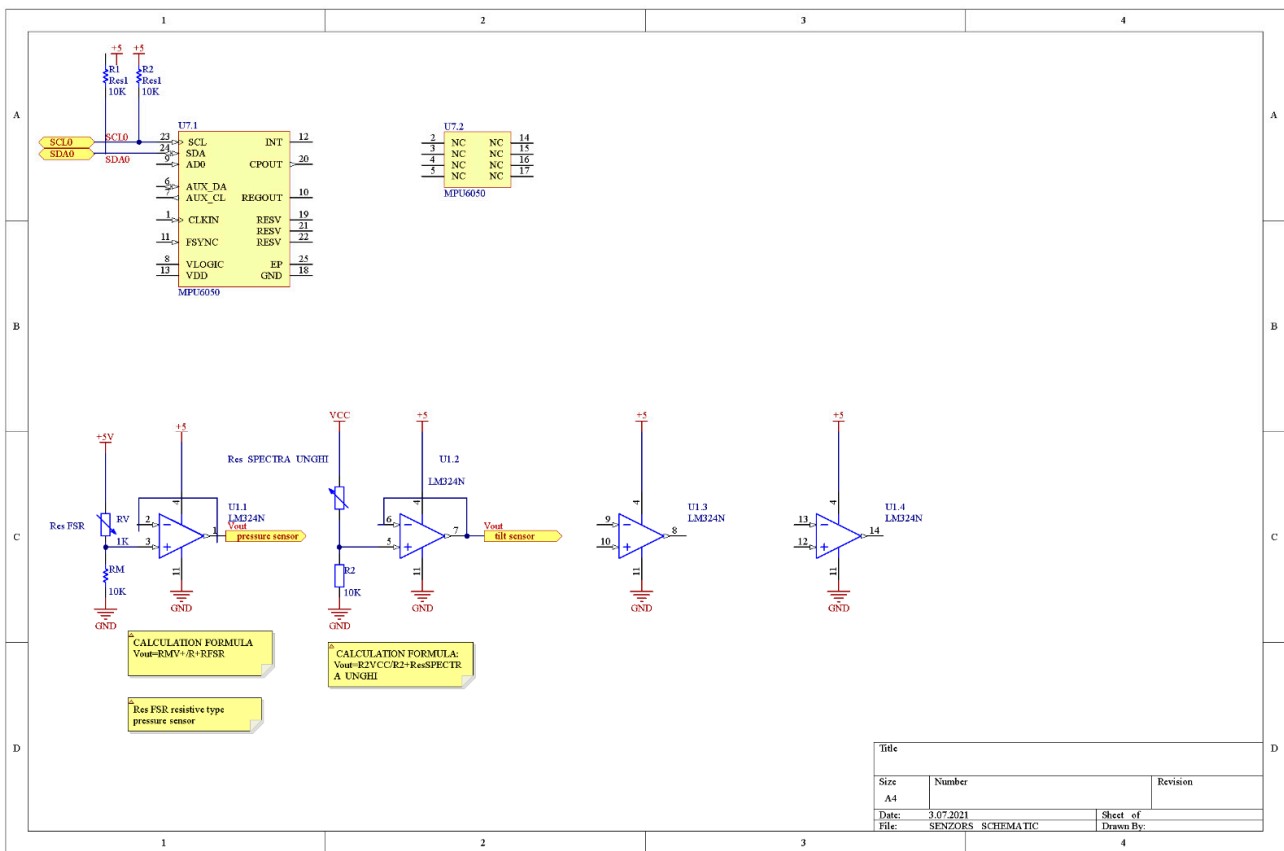

**Figure A2.** Sensors schematic.

## Appendix B

```
test__presiune__v | Arduino 1.8.14 Hourly Build 2020/10/09 12:33
File  Edit  Sketch  Tools  Help

test__presiune__v

//soft  testare  senzor  presiune  FSR400  INTERLINK
int fsrPin = 0; // the FSR and 10K pulldown are connected to a0
int fsrReading; // the analog reading from the FSR resistor divider
int fsrVoltage; // the analog reading converted to voltage
unsigned long fsrResistance; // The voltage converted to resistance, can be very big so make "long"
unsigned long fsrConductance;
long fsrForce; // Finally, the resistance converted to force
void setup(void) {
Serial.begin(9600); // We'll send debugging information via the Serial monitor
}
void loop(void) {
fsrReading = analogRead(fsrPin);
Serial.print("Analog reading = ");
Serial.println(fsrReading);
// analog voltage reading ranges from about 0 to 1023 which maps to 0V to 5V (= 5000mV)
fsrVoltage = map(fsrReading, 0, 1023, 0, 5000);
Serial.print("Voltage reading in mV = ");
Serial.println(fsrVoltage);
Serial.println(fsrVoltage);
if (fsrVoltage == 0) {
Serial.println("No pressure");
} else {
// The voltage = Vcc * R / (R + FSR) where R = 10K and Vcc = 5V
// so FSR = ((Vcc - V) * R) / V yay math!
fsrResistance = 5000 - fsrVoltage; // fsrVoltage is in millivolts so 5V = 5000mV
fsrResistance *= 10000; // 10K resistor
fsrResistance /= fsrVoltage;
Serial.print("FSR resistance in ohms = ");
Serial.println(fsrResistance);
fsrConductance = 1000000; // we measure in micromhos so
fsrConductance /= fsrResistance;
Serial.print("Conductance in microMhos: ");
Serial.println(fsrConductance);
// Use the two FSR guide graphs to approximate the force
if (fsrConductance <= 1000) {
fsrForce = fsrConductance / 80;
Serial.print("Force in Newtons: ");
Serial.println(fsrForce);
} else {
fsrForce = fsrConductance - 1000;
fsrForce /= 30;
Serial.print("Force in Newtons: ");
Serial.println(fsrForce);
}
}
Serial.println("TIME   FOR DELAY 1000  ");
delay(1000);
}
```

**Figure A3.** *Cont.*

**Figure A3.** Application software for gripping force measurement.

```
SENZOR_INCLINARE_V | Arduino 1.8.14 Hourly Build 2020/10/09 12:33
File  Edit  Sketch  Tools  Help

SENZOR_INCLINARE_V

//MASURAREA INCLINARII
//Create a voltage divider circuit combining a flex sensor with a 47k resistor.
//- The resistor should connect from A0 to GND.
//- The flex sensor should connect from A0 to 3.3V
//As the resistance of the flex sensor increases (meaning it's being bent), the
//voltage at A0 should decrease.

//Development environment specifics:
//Arduino 1.8.14

const int FLEX_PIN = A0; // Pin connected to voltage divider output

// Measure the voltage at 5V and the actual resistance of your
// 47k resistor, and enter them below:
const float VCC = 4.98; // Measured voltage of Ardunio 5V line
const float R_DIV = 47500.0; // Measured resistance of 3.3k resistor

// Upload the code, then try to adjust these values to more
// accurately calculate bend degree.
const float STRAIGHT_RESISTANCE = 31000.0; // resistance when straight
const float BEND_RESISTANCE = 90000.0; // resistance at 90 deg

void setup()
{
  Serial.begin(9600);
  pinMode(FLEX_PIN, INPUT);
}

void loop()
{
  // Read the ADC, and calculate voltage and resistance from it
  int flexADC = analogRead(FLEX_PIN);
  float flexV = flexADC * VCC / 1023.0;
  float flexR = R_DIV * (VCC / flexV - 1.0);
  Serial.println("Resistance: " + String(flexR) + " ohms");

  // Use the calculated resistance to estimate the sensor's
  // bend angle:
  float angle = map(flexR, STRAIGHT_RESISTANCE, BEND_RESISTANCE,
                    0, 90.0);
  Serial.println("Bend: " + String(angle) + " degrees");
  Serial.println();

  delay(500);
}
```

```
Done uploading.
Sketch uses 6452 bytes (20%) of program storage space. Maximum is 32256 bytes.
Global variables use 232 bytes (11%) of dynamic memory, leaving 1816 bytes for lo
Invalid library found in C:\Users\harnicul\Documents\Arduino\libraries\fsr__test:
```

**Figure A4.** *Cont.*

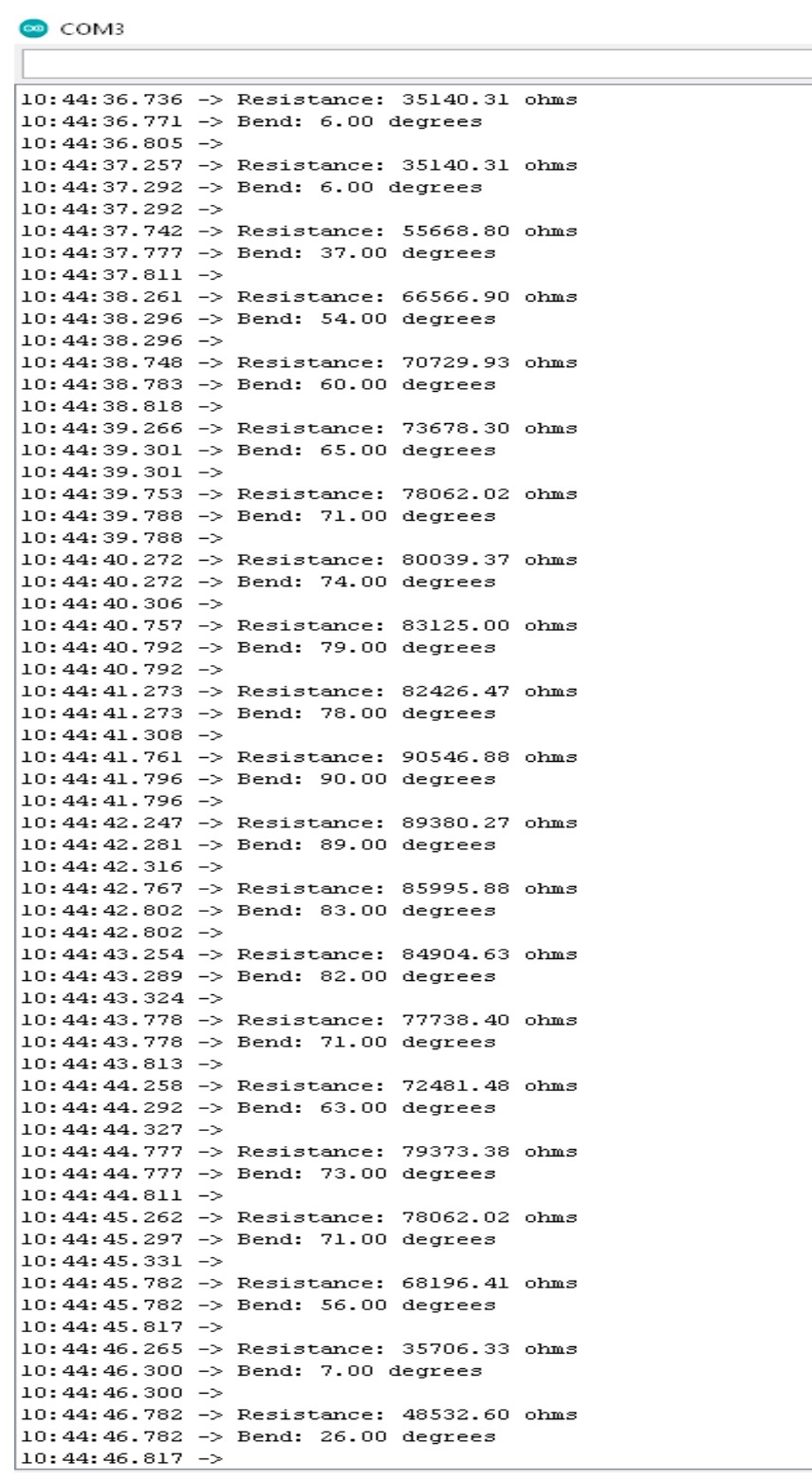

**Figure A4.** Application software for tilt angle measurement.

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
