# Peer review of "Concept, Design, Initial Tests and Prototype of Customized Upper Limb Prosthesis"

_applsci, doi:10.3390/app11073077_

Round 1

Reviewer 1 Report

The presented article is about the development of a myoelectric upper limb prosthesis. This is rather a development documentation than a scientific paper. There are many mistakes in the editing and the structure of the article. 

  1. Punctuation: Many points, colons, semicolons are missing. Example:
    "
Link – is a rigid body. 88
Joint – is a permanent contact between two links 89
Planar motion – is the motion “in which all points belonging to a link move in a 90
plane known as the plane of motion, while simultaneously the link is free to rotate about 91
an axis perpendicular to the plane of motion”

Is it a list? Where are the sentence closing points? If this is a list then itemize them. All sentences starts now in a new paragraph.  Please check all sentences. 

2. ALL equations are not aligned to the text and the equation numbers. It seems very ugly.

3. Many figures are splited into two sides. Those are not required. 

4. There are many wide Figures, they are more wider than the width of the paragraph. See Figures 1, 2, 3,4,5,6.....etc

5. Figure 1 contains two splited parts. But on the first one there is still a "c".

6. in the figures, the notations are not always explained

7. Figure 1. c  barely visible 

8. Many equations are wider than the paragraph. See eq 3, 6, 10, 12

9. PLease correct the english!

10. Usually the variable are italics during the text. I mean: "restricting i degrees of freedom"

11. the variable with lower index: please use subscripts. There are many-many variables as: ai, Ti, Ti+1,..etc. Please use equation editor or subscripts not small characters.  In lines 149 and 150 there is a Ti and a small Ti+1. 

12. There is a list 170-172. This is not itemized and -as I wrote before- there are many small  variables: 

 s3 = 0; s4 = 0; s5 = 0 and  a1 = 0; a2 = 0; a5 = 0. Check them as subscripts. 

13. That is not defined how the dimensions and dimension parameters are resulted from the scanning

14. Figure 5 writes about PRESSURE and TEMPERATURE control, but what is the pressure and temperature here- not defined.  Otherwise Fig. 5. is unnecessary

15. Fig 6 is so meaningless and poorly edited. The ADC is written in two lines. Why? This is a very bascis figure and there are many modules that are missing (signal processing, signal conditioning( amplifying?), motor controlling. 

Is there feedback (torque) from the motor? If yes please put it here

16.  Line 217: We still don't know why resistive and temperature sensors are used

17. MyoWare not MayoWare!!!!!!

17. Fig. 7: very basic, coming from the MyoWare manual. unnecessary.

18. Fig 7 and 8 can not be seen, very tiny, meaningless . This is rather electrical engineering. This is scanned with light yellow background. This is not good. 

19. Fig 9 contains a MATLAB script with magic parameters and without explanation. This is a calculation only, remove it. 

20. Line 261: Qi: "i" must be a subscript. I see Q-s only in the codes. 

21. Line 262: Where did the 20N come from? 

22. Fig. 14: Basic calculations, very wide, unnecessary

23: Fig. 15: Why is the table with the same values?

24: Fig. 16  occupies an entire page. Reduce the number of the design sub-figures. 

25: Line 308: "Furher"....

26: Line 313: Please correct the sentences as: "Some preliminary experiments, so that to test the accuracy of sensors were done."

27: Fig. 18: unnecessary

28: If the dimensions of the 3D printed parts are measures then you have to put the statistics. Without it a measurement doesn’t make much sense.

29. Fig 23  is not  aligned (center). It is written "Then the downface of the components are calculated on the premise of angle and slope as can be seen in the Figure 23." It cannot be seen from Fig. 23 and I don't understand the meaning of the Figure

30. Figures 23, 24, 25 have dark (black) background. Cannot be seen. They contains very small parts they are not visible. 

31. The discussion is very short. 

Author Response

The reviewer’s comments, point-by-point response, are shown in the Attached file.

Reviewer 2 Report

The Corina Radu’s et al, have done very good work in the manuscript " Concept, design, initial tests and prototype of customized upper limb prosthesis ".

The manuscript has been well written, methodology elaborately discussed, results are clearly described, sound to readers, and categorized very well.

Author Response

The authors do really appreciate reviewer’s comments.

Kind regards.

Reviewer 3 Report

Review of applsci-1120809-peer-review-v1

Concept, design, initial tests and prototype of customized upper limb prosthesis

This study describes XXX

Please see my comments below.

General:

Important comment: The language is not good so that some sentences and meanings are lost to the reader. The manuscript should go through a thorough linguistic editing. I tried my best to understand what the authors tried to convey but I admit that some parts were unintelligible.

Abstract:

  • It is not clear from the abstract, what was the rational for the study (what is the gap in the literature?), what was the aim of the study? The tools are partly explained and need further details (what sensors? Pressure? Was this a myoelectric hand? More…), what are the results?  Did you test it on subjects? If not, what were the variables for validation?  Quantification of results is needed.

Introduction:

  • The first paragraph should focus on upper extremity amputees (statistics, quality of life etc) with relevant references from the literature.
  • "The artificial hands presently in use are complicated in design and control structure" – I do not agree. Mechanical prosthesis are quite simple. Maybe you are referring to myoelectric prostheses?
  • The anatomical description of the human hand is redundant.
  • The rational for the study is extremely weak, mainly due to the lacking literature review. Relevant papers that presented the concept, design and prototype for a customized low cost upper limb prosthesis (the study aim) exists but were not mentioned herein. See some recent examples below. The authors should state the main novelty of their work in relation to existing literature.
    • Lee KH, Bin H, Kim K, Ahn SY, Kim BO, Bok SK. Hand Functions of Myoelectric and 3D-Printed Pressure-Sensored Prosthetics: A Comparative Study. Ann Rehabil Med. 2017 Oct;41(5):875-880. Do     i: 10.5535/arm.2017.41.5.875. Epub 2017 Oct 31. PMID: 29201828; PMCID: PMC5698676.
    • Roland T, Amsuess S, Russold MF, Baumgartner W. Ultra-Low-Power Digital Filtering for Insulated EMG Sensing. Sensors (Basel). 2019 Feb 24;19(4):959. doi: 10.3390/s19040959. PMID: 30813494; PMCID: PMC6412999.
    • Prakash A, Kumari B, Sharma S. A low-cost, wearable sEMG sensor for upper limb prosthetic application. J Med Eng Technol. 2019 May;43(4):235-247. doi: 10.1080/03091902.2019.1653391. Epub 2019 Aug 15. PMID: 31414614.
    • Ku I, Lee GK, Park CY, Lee J, Jeong E. Clinical outcomes of a low-cost single-channel myoelectric-interface three-dimensional hand prosthesis. Arch Plast Surg. 2019 Jul;46(4):303-310. doi: 10.5999/aps.2018.01375. Epub 2019 Jul 15. Erratum in: Arch Plast Surg. 2019 Sep;46(5):491. PMID: 31336417; PMCID: PMC6657188.
    • Ku I, Lee GK, Park CY, Lee J, Jeong E. Clinical outcomes of a low-cost single-channel myoelectric-interface three-dimensional hand prosthesis. Arch Plast Surg. 2019 Sep;46(5):491. doi: 10.5999/aps.2019.01256. Epub 2019 Sep 15. Erratum for: Arch Plast Surg. 2019 Jul;46(4):303-310. PMID: 31550753; PMCID: PMC6759443.
    • Wu H, Dyson M, Nazarpour K. Arduino-Based Myoelectric Control: Towards Longitudinal Study of Prosthesis Use. Sensors (Basel). 2021 Jan 24;21(3):763. doi: 10.3390/s21030763. PMID: 33498801; PMCID: PMC7866037.

Methods:

  • This section is filled with redundant explanations of basic concepts (DOF, planar motion, kinematic analysis etc.). This should be omitted as it is basic textbook knowledge or moved to an appendix with references to quoted explanations.
  • Fig. 1: Please explain each notation shown in the figure (what are M, G, F, etc?). What are the references of the Figures?  Were the figures replicated from another paper or are these original (if so, then again, why the references…)?
  • Also for fig. 2. What do the notations stand for?  I am not sure what I am looking at. Please provide exact details.
  • As with the concepts of DOF etc, most of the equations are also basic, e.g. the transformation matrix…. Did the authors made any changes to these models?  What is the novelty here?
  • "The mechatronic system for the upper limb has been designed so that to enable fingers and arm motions as much as possible like the ones of a real limb" – how? What was the ROM of each joint? What reference di you use for it?
  • Please provide details regarding the subject: was ethical approval granted from an IRB? Male/female? Age? Left/right hand?
  • Please provide further details regarding the scanning: scanner resolution? Did you clean the scanned image? If so, what was the cleaning process?
  • "real upper limb" – physical one? As opposed to virtual one?
  • Slicing: how many planes? What was the distance between planes?
  • Lines 195-200: these are neither tools nor methods.
  • The algorithm in Fig. 5 is the one used to activate and control some of the available commercial myoelectric prostheses. How is this one new?
  • "sensors signal" – what signal is that?  Temperature? Pressure? Vibration?
  • What is the properties of the EMG sensor (SNR etc….)?
  • Fig. 7: fonts are too small. It is unreadable. Again, this information is quite redundant. I do not see any novelty in the electronic circuits. Same for the Matlab output – why do you show it? If this is somehow important, you have to explain each line and notation (for example, what is th210?.....)

Results:

  • "As remark, we could assess that there has been a lot of work, many constraints to 296 overcome and a hard task to design the right, innovative and accurate mechatronic sys-297 tem for the upper limb prosthesis. Still, improvements are going to be done, after differ-298 ent experiments and tests." – this is an important note. I would recommend that the algorithm be tested and validated before being published.
  • How is the process of manufacturing part of the results?

Discussion:

  • The discussion and conclusions are not really relevant to most of the results. There was no actual testing, no quantitative parameters, I am not even sure that there was a set aim to the study. Also, there is no actual novelty when comparing the outcomes of this work and available commercial myoelectric prostheses.

Author Response

The reviewer’s comments, point-by-point response, are shown in the attached file.

Round 2

Reviewer 1 Report

The paper is better than before, there only a few things that must be correccted:

Figure 1: (a) and (b) blurred, these quality is worse than the (c) and cannot see the text on subfigure (a).

 The variable subscripts are not subscripts but also a small letter (as Ti in line 347, T1 355, 445,...etc). Check the entire document.

Reviewer 3 Report

Please find attached review of the submission. I still believe this paper should be rejected for the following reasons:

1) They performed scans on a subject, without obtaining an ethics approval (most journals have a filter that during submission, if there is no ethics approval and human subjects were recruited, then it is rejected by the editor).

2) There is no actual novelty here.

3) The presentation is extremely bad.